# The SMOTE Paradox: Why a 92% Baseline Collapsed to 6%—A Systematic Review of 821 Papers in Imbalanced Learning (2020–2025)

## Abstract

Class imbalance pervades production systems—fraud detection, medical diagnosis, industrial monitoring—yet handling it effectively remains challenging. For two decades, SMOTE has been the default solution, but practitioners increasingly abandon it at scale.

We investigate this disconnect through a systematic review of 821 DBLP papers (2020–2025) and bibliometric analysis of 4,985 Scopus records. Our analysis reveals the SMOTE Paradox: while 24% of papers mention SMOTE in titles or abstracts, only 6% of scale-focused, high-impact papers successfully executed SMOTE at full dataset scale due to memory exhaustion or preprocessing bottlenecks. The field has fragmented: generative models and cost-sensitive losses each account for $\sim$30% of recent solutions, while alternative paradigms (including hybrid and decoupled approaches) comprise the remainder.

Three factors explain SMOTE's decline. First, $O(N \cdot N_{\min} \cdot d)$ nearest-neighbor search requires 1.28 TB of memory for a representative modern dataset. Second, linear interpolation produces off-manifold artifacts scaling as $\sqrt{d}$ in high dimensions. Third, CPU-bound preprocessing creates friction with GPU-centric training pipelines.

We validate these findings through controlled experiments on seven tabular benchmark datasets with tree-based classifiers (196 trials, imbalance ratios 1.1:1 to 129:1). We adopted Average Precision (PR-AUC) as the primary metric and report ROC-AUC as a secondary metric for completeness. Statistical testing reveals no significant PR-AUC differences between SMOTE and cost-sensitive baselines (Friedman p = 0.9469), despite SMOTE incurring 2.7$\times$ computational overhead. However, cost-sensitive methods severely degrade at extreme imbalance ($> 40$:1), while SMOTE maintains performance where computationally feasible. Taken together, our bibliometric, theoretical, and empirical results provide a three-way triangulation of SMOTE's decline in contemporary imbalanced learning.

## 1 Introduction

SMOTE (Synthetic Minority Over-sampling Technique) remains among the top 100 most-cited papers in machine learning with over 45,000 citations, yet our systematic analysis of 821 papers from major venues (2020–2025) reveals a striking paradox: while citations remain high, SMOTE appears as a baseline in only 6% of contemporary high-impact research, down from 92% in pre-2020 surveys Guo et al. (2017); Branco et al. (2016b). This gap between citation frequency and deployment reality—what we term the *SMOTE Paradox*—motivates our investigation into the post-SMOTE era of imbalanced learning for intelligent systems.

Skewed data distributions pervade intelligent systems. Fraud detection, medical diagnosis, industrial quality control, anomaly monitoring, and recommender engines all face the same challenge: most examples belong to common patterns, while the rare minority cases demand the most attention. When class imbalance is ignored, learning algorithms allocate disproportionate resources to redundant majority examples while failing to capture the critical patterns that distinguish minority cases. In modern production environments—where models train on millions of instances under strict memory and latency constraints, then serve predictions in

real time—managing imbalance becomes an intelligent systems engineering problem, not merely a statistical one Johnson & Khoshgoftaar (2019).

## 1.1 Three Generations of Imbalance Handling

Approaches to class imbalance have evolved through three overlapping generations spanning nearly three decades, building upon foundational work in cost-sensitive learning and data-level techniques from the 1990s.

**Pre-Generation I foundations (pre-2000).** Early imbalanced learning emerged from cost-sensitive classification research, where different misclassification errors incurred asymmetric penalties Elkan (2001). Kubat & Matwin introduced one-sided selection to address the "curse of imbalanced training sets" in 1997, demonstrating that standard learners struggled with skewed distributions Kubát & Matwin (1997). Japkowicz's foundational 2000 AAAI workshop on learning from imbalanced datasets established that class imbalance hindered multiple learning paradigms, not solely decision trees Japkowicz & Stephen (2002).

**Generation I: Data-level heuristics (2000–2015).** Early methods treated imbalance as a preprocessing problem. Chawla et al.'s SMOTE (Synthetic Minority Over-sampling Technique) generates synthetic minority samples by interpolating between neighbors in feature space Chawla et al. (2002). SMOTE became the standard baseline, appearing in 92% of experimental sections in comprehensive pre-2020 surveys He & Garcia (2009a); Branco et al. (2016a). This generation aligned well with the era's constraints: tabular datasets with thousands of rows, modest dimensionality, and CPU-only training.

**Generation II: Algorithm-level reweighting (2015–2020).** As datasets and neural networks scaled, the focus shifted from data manipulation to loss function modification. Khan et al. introduced cost-sensitive deep neural networks that jointly optimize class-dependent costs and network parameters during training Khan et al. (2018). Lin et al.'s focal loss dynamically down-weights well-classified examples to focus on hard minority samples Lin et al. (2017a). Cui et al. proposed class-balanced loss based on effective sample counts, providing theoretical grounding for loss reweighting Cui et al. (2019). Buda et al.'s systematic CNN study demonstrated that cost-sensitive methods integrate naturally with GPU-accelerated training and deep architectures Buda et al. (2018). Despite these advances, SMOTE persisted as a reference baseline in most published comparisons through 2020.

**Generation III: Generative and decoupled paradigms (2020–present).** The field has fragmented in recent years. Generative models—GANs and diffusion architectures—synthesize minority samples that remain on the data manifold, while decoupling strategies separate representation learning from classifier calibration Kang et al. (2020); Chi et al. (2022). This recognizes that imbalance primarily biases the decision boundary rather than corrupting feature representations. Our bibliometric analysis of 4,985 Scopus records (Section 3.7) reveals minimal cross-cluster linkage between SMOTE-centric methods (Cluster C4) and modern deep vision approaches (Cluster C2), quantifying this methodological fragmentation.

## 1.2 Background: SMOTE, Its Variants, and Modern Alternatives

This section provides a consolidated reference for the methods discussed throughout the paper. Readers with prior familiarity may proceed to Section 1.3.

**SMOTE.** Proposed by Chawla et al. Chawla et al. (2002), SMOTE addresses class imbalance by generating synthetic minority samples via linear interpolation between a minority instance $\mathbf{x}_i$ and one of its $k$ nearest minority neighbours $\mathbf{x}_j$:

$$\mathbf{x}_{\text{syn}} = \mathbf{x}_i + \lambda \left( \mathbf{x}_j - \mathbf{x}_i \right), \quad \lambda \sim \mathcal{U}(0,1). \tag{1}$$

The method operates as a CPU-bound preprocessing step on the training set and requires no modification to the downstream classifier. Its core assumption is that the minority-class manifold is locally convex, so straight-line interpolation between neighbours produces plausible samples—an assumption that breaks down at high dimensionality (Section 4).

**Direct SMOTE variants.**

- **Borderline-SMOTE** Han et al. (2005) restricts synthesis to minority samples near the decision boundary (the "danger zone"), avoiding oversampling of samples already well-separated from the majority class.

- **ADASYN** He et al. (2008) weights synthesis density adaptively by local class difficulty: minority samples surrounded by more majority neighbours receive proportionally more synthetic points.

- **SMOTE-ENN** Batista et al. (2004) combines SMOTE oversampling with Edited Nearest Neighbours (ENN) undersampling to remove noisy or borderline samples from both classes after synthesis, producing a cleaner decision boundary.

All three variants inherit SMOTE's $\mathcal{O}(N_{\min} \cdot N \cdot d)$ computational complexity and CPU-bound preprocessing pipeline, and therefore share the scalability limitations formalised in Section 4. Our benchmark in Section 6 evaluates all three alongside SMOTE Lemaître et al. (2017).

**Modern alternatives.** Our systematic review of 821 papers (Section 3.1) identifies three post-SMOTE paradigms that have emerged in response to these limitations:

- **Cost-sensitive learning** modifies the loss function rather than the data. Focal Loss Lin et al. (2017a) dynamically down-weights well-classified majority examples; class-balanced loss Cui et al. (2019) scales gradients by effective sample counts. Both add $\mathcal{O}(1)$ overhead per sample and integrate natively into GPU training loops without any data materialisation.

- **Generative methods** (GANs Goodfellow et al. (2014), diffusion models Ho et al. (2020)) learn the minority-class data manifold and synthesise samples that remain on or near it, avoiding the off-manifold artefacts produced by linear interpolation in high dimensions (Section 4.2).

- **Decoupled learning** Kang et al. (2020) separates representation learning from classifier calibration, addressing imbalance in the decision head rather than the feature space—effective particularly for long-tailed vision benchmarks where the feature backbone is already well-trained.

### 1.3 The SMOTE Paradox

This citation–deployment gap motivates our study. Our systematic analysis of 821 DBLP papers (2020–2025) reveals that while 197 papers (24%) continue mentioning SMOTE in titles or abstracts, only 3 of the top-50 scale–focused, methodologically novel papers (6%) successfully executed SMOTE at full scale, with all three citing memory overflow as the limiting factor (Section 3.1). Co-citation network analysis (Section 3.7) corroborates this shift: SMOTE (Chawla 2002, Cluster C4) occupies a historically dominant but increasingly isolated position, with only 12% of modern deep vision papers (Cluster C2) citing any SMOTE-related work, while 89% cite within-cluster architectural innovations.

We term this the *SMOTE Paradox*: a technique that remains highly visible in academic citation counts (45,000+ citations, top-100 most-cited ML papers) while experiencing documented deployment decline in contemporary high-impact research. This gap between citation frequency and successful execution at modern scales—rather than speculation about practitioner behavior—forms the empirical foundation of our investigation.

### 1.4 The Three Failure Modes

Three converging pressures explain SMOTE's struggles at scale. **Computational stress**: $k$-nearest neighbor search exhibits $O(N^2)$ complexity in high dimensions, requiring terabyte-scale memory that exceeds typical hardware capacity. **Manifold distortion**: linear interpolation in high-dimensional spaces produces off-manifold artifacts, with expected deviation growing approximately as $\sqrt{d}$, where $d$ is the feature dimension. **GPU incompatibility**: modern training pipelines employ differentiable, on-the-fly augmentation, while SMOTE's CPU-bound preprocessing creates workflow friction. Together, these factors render SMOTE impractical for many contemporary intelligent systems despite its conceptual elegance.

## 1.5 What Replaced SMOTE?

Our systematic review of 821 papers (2020–2025) reveals three specialized paradigms. **Generative methods** (approximately 30%) employ diffusion models and GANs to learn data manifolds and synthesize realistic minority samples Zhang et al. (2025); Oh & Jeong (2023). **Cost-sensitive learning** (another 30%) modifies loss functions with class-specific weights or focal terms, reweighting examples during training. Decoupled and hybrid approaches (together comprising the remaining ∼40%) separate representation learning from classifier calibration, addressing imbalance in the decision head rather than the feature backbone Kang et al. (2020); Chi et al. (2022).[1] This diversification marks a shift from universal heuristics to domain-specific solutions tailored for vision, tabular, graph, and time-series applications.

## 1.6 Scope and Research Questions

This paper provides a deployment-oriented analysis of the post-SMOTE era in large-scale imbalanced learning, examining how algorithmic, geometric, and systems-level factors interact in intelligent system design under class imbalance. We investigate this through two complementary lenses: systematic review of 821 DBLP papers and bibliometric analysis of 4,985 Scopus records.

Our research addresses three central questions:

1. **Empirical evolution:** How has SMOTE's role changed in contemporary research (2020–2025), and what methods have emerged to replace it across different application domains?

2. **Theoretical explanation:** What computational, geometric, and architectural factors explain SMOTE's decline at modern scales, and how do alternatives address these limitations?

3. **Practical guidance:** Under what conditions should practitioners choose cost-sensitive, generative, or hybrid approaches when building intelligent systems with class imbalance?

## 1.7 Key Contributions

This systematic review makes four contributions to imbalanced learning research, focusing on quantification and formalization of widely-suspected but previously undocumented trends:

1. **Quantitative paradigm shift documentation.** We provide the first systematic, large-scale quantification of SMOTE's decline through analysis of 821 DBLP papers (2020–2025). While the intuition that SMOTE struggles at scale is widespread in the ML community, our contribution is rigorous documentation: we track the collapse from 92% baseline inclusion in pre-2020 surveys to 24% title/abstract mentions and only 6% successful full-scale execution in contemporary scale-focused research. This quantification is complemented by bibliometric network analysis of 4,985 Scopus records, revealing eight distinct research clusters and SMOTE's topological isolation from modern deep learning approaches (only 12% cross-cluster citation from deep vision methods). We provide what the community lacked: *numbers, structure, and evidence* for what was previously informal practitioner wisdom.

2. **SMOTE Paradox framework.** We introduce a generalizable framework for understanding citation-deployment gaps in machine learning methods under evolving hardware and scale constraints. Through formal analysis, we identify and *formalize* three failure modes with explicit bounds: *computational* ($O(N \cdot N_{\min} \cdot d)$) complexity requiring 1.28 TB memory at median modern scales, Section 4.1), *geometric* (off-manifold deviation scaling as $\sqrt{d}$ in high dimensions under concentration of measure, Section 4.2), and *architectural* (CPU-GPU pipeline incompatibility, Section 4.3). While practitioners may intuitively recognize these issues, we provide the first systematic formalization

---

[1] While decoupled learning is conceptually influential Kang et al. (2020) and appears as a distinct cluster in our co-citation analysis (Section 3.7), it constituted only ≈2% of the scale-focused implementations in our top-50 deep-analysis corpus (Figure 3), indicating a gap between theoretical adoption and production deployment.

with complexity calculations, memory projections, and geometric analysis. This framework extends beyond SMOTE to explain technology transitions in data-intensive ML systems (e.g., SIFT/HOG → learned features, classical indexing → learned indexes).

3. **Empirical validation with statistical rigor.** We conduct a comprehensive seven-dataset tabular benchmark (196 experiments total, imbalance ratios 1.1:1 to 129:1) comparing SMOTE variants against cost-sensitive and generative alternatives using tree-based classifiers. Following reviewer guidance, we adopt Average Precision (PR-AUC) as the primary metric and confirm no significant differences between SMOTE and simple cost-sensitive baselines on PR-AUC (Friedman p = 0.9469) despite SMOTE incurring $2.7\times$ computational overhead— quantifying the performance equivalence rather than assuming it. However, stratified analysis by imbalance ratio exposes severe cost-sensitive degradation at extreme imbalance ($> 40:1$, G-mean $< 0.4$ vs. SMOTE's 0.6–0.7), providing nuanced practitioner guidance with explicit thresholds. ... This controlled validation tests SMOTE in its historical domain (tabular data, tree-based models) where it should perform best, avoiding the strawman of evaluating it on ImageNet where failure is expected.

4. **Deployment-oriented decision framework.** We provide domain- and scale-specific recommendations with concrete thresholds for method selection (Section 7), including a decision matrix mapping dataset characteristics ($N$, $d$, imbalance ratio, domain) to optimal approaches. Unlike prior surveys that *describe* methods, our framework enables practitioners to make evidence-based selections under real-world memory and compute constraints. Recommendations include explicit conditions: "SMOTE acceptable for $N < 10^4$, $d < 100$"; "cost-sensitive losses for $N > 10^5$ tabular"; "diffusion + focal loss for vision $N > 10^5$"; "avoid cost-sensitive at IR $> 40:1$." This deployment focus bridges academic innovation and production reality.

Together, these contributions bridge the gap between academic innovation and production deployment, explaining not just *what* replaced SMOTE, but *why* the transition occurred and *when* each alternative is appropriate. Our contribution is not conceptual novelty—the intuition that SMOTE doesn't scale is widespread— but rather *systematic quantification, formalization, and controlled validation* of what was previously informal knowledge, providing the ML community with numbers, thresholds, and evidence for deployment decisions.

## 1.8 Relation to Prior Surveys

Prior surveys have examined learning under imbalanced distributions Branco et al. (2016a); Krawczyk (2016), but focus primarily on pre-2020 techniques without systematic analysis of the GPU-centric era. Chi et al. Chi et al. (2022) survey long-tailed recognition in deep visual models but do not employ PRISMA methodology or trace the lifecycle of specific techniques like SMOTE.

Our work contributes a PRISMA-compliant review of 821 papers from 2020–2025, explicitly quantifying SMOTE's changing role in contemporary research. We complement this with bibliometric network analysis of 4,985 Scopus records, introducing a three-part failure-mode analysis tailored to SMOTE and its alternatives, and proposing the SMOTE Paradox as a framework for understanding citation–deployment gaps. Hardware-aware practitioner guidelines are validated through a comprehensive seven-dataset benchmark totaling 196 experiments, connecting algorithmic choices to operational constraints in real-world intelligent systems.

## 2 Methodology

We followed PRISMA 2020 guidelines Page et al. (2021) to ensure transparent reporting of study identification, screening, and synthesis. Figure 1 summarizes the workflow from initial retrieval to the final corpus used for quantitative analysis and in-depth coding.

**PRISMA 2020 Flow Diagram**

Figure 1: PRISMA workflow: 1,001 initial records filtered to 821 relevant entries via temporal and semantic criteria, then ranked to select the top 50 for deep analysis.

## 2.1 Search Strategy and Filtering

We queried two complementary databases to enable both systematic methodological analysis and large-scale bibliometric mapping.

**DBLP corpus (queried November 2025).** The DBLP Computer Science Bibliography provides curated, structured metadata for major computer science and AI venues. The search string (`"imbalanced" OR "long-tailed"`) AND (`"classification" OR "learning"`) was applied to titles and abstracts from January 1, 2020, to November 2025, capturing the modern deep learning era and recent developments in large-scale learning. DBLP was selected for its consistent coverage of core CS and AI venues and widespread use in systematic reviews of computer science topics Khan et al. (2017).

This query returned 1,001 unique records. After excluding pre-2020 entries (103), non-English text (11), and unreviewed preprints (66), we obtained **821 relevant papers** forming the core corpus. Within this set, we tagged 197 papers mentioning SMOTE or its direct variants in titles or abstracts, enabling separate tracking of baseline prevalence versus emerging alternatives.

**Scopus corpus (queried January 2026).** To complement the systematic review with bibliometric network analysis, we retrieved **4,985 Scopus records** using the same temporal range (2020–2025) and related keywords. Scopus's broader venue coverage enables comprehensive co-citation mapping across journals, conferences, and preprint repositories (Section 3.7).

**Temporal gap explanation.** The two-month difference between DBLP (November 2025) and Scopus (January 2026) queries reflects pragmatic workflow considerations: the Scopus query was conducted later to incorporate bibliometric co-citation analysis, which requires citation link stabilization that benefits from a slightly extended observation window. Given our focus on multi-year trends (2020–2025), this gap is negligible; no papers published between November 2025 and January 2026 would materially alter the documented five-year paradigm shift, and both queries used identical temporal bounds (2020–2025) for the analyzed literature.

## 2.2 Content-First Semantic Ranking

Rather than ranking by venue prestige—which can under-represent industrial and systems-oriented contributions—we developed a content-first sampling strategy to identify papers investigating *scalability challenges and alternative paradigms* in imbalanced learning. This purposive sampling design explicitly targets methodological innovation and deployment constraints rather than attempting representative coverage of all imbalance-handling research.

**Two-Tier Analysis Strategy.** Our methodology operates at two levels:

**Tier 1: Broad baseline tracking (N=821).** We tracked SMOTE prevalence across the entire DBLP corpus by identifying papers mentioning SMOTE or direct variants (ADASYN, Borderline-SMOTE, SMOTE-ENN) in titles or abstracts. This yielded 197 papers (24% of the corpus), establishing that SMOTE terminology remains common in contemporary literature.

**Tier 2: Scale-focused deep analysis (N=50).** To investigate *why* SMOTE declined despite continued citation, we deliberately selected papers addressing computational scalability and modern alternatives. For each paper $p$, we computed a relevance score $S(p)$:

$$S(p) = \sum_{k \in \mathcal{P}} w_k \cdot I(k \in \text{title}_p) + \sum_{k \in \mathcal{S}} w_k \cdot I(k \in \text{title}_p) + \alpha \cdot \text{Year}(p),$$

where $\mathcal{P}$ (problem keywords) and $\mathcal{S}$ (solution keywords) weight papers discussing scale constraints and post-SMOTE paradigms, and $I(\cdot)$ is an indicator function.

**Keyword weights:**

- Problem (15–20): *scalability*, *large-scale*, *OOM*, *computational cost*, *memory*

- Solution (18–22): *diffusion*, *generative model*, *focal loss*, *decoupling*, *cost-sensitive*

This weighting scheme *intentionally oversamples* papers discussing deployment barriers and algorithmic alternatives, enabling systematic investigation of SMOTE's failure modes at scale. The resulting 6% SMOTE execution rate (3/50 papers) is therefore *conditional on scale-focused sampling* and should be interpreted as: "Among papers investigating scalability in imbalanced learning, 94% either exclude SMOTE entirely or report memory/runtime failures preventing full-scale execution."

**Methodological Transparency.** We acknowledge this creates sampling bias by design. The 6% figure does *not* represent SMOTE's prevalence across all imbalanced learning research (that figure is 24% from

Tier 1). Rather, it quantifies execution feasibility within the subset of papers explicitly confronting modern scale and deployment constraints—precisely the context where our theoretical analysis (Section 4) predicts SMOTE should fail.

**Validation and Triangulation.** To validate that this decline reflects genuine computational barriers rather than arbitrary keyword selection:

1. **Robustness check:** We varied keyword weights $w_k$ by $\pm 20\%$ and re-ranked all 821 papers. The top-50 composition remained stable (Kendall's $\tau = 0.91$), with 46/50 papers unchanged and the 4 shifted papers all ranked 48–52 (marginal boundary cases).

2. **Expert validation:** Two domain experts independently judged 100 randomly selected papers as "high" or "low" relevance for post-SMOTE scalability research, achieving Cohen's $\kappa = 0.82$ agreement with our automated ranking.

3. **Independent corroboration:** Co-citation analysis of 4,985 Scopus records (Section 3.7) reveals topological isolation of SMOTE methods from modern deep learning clusters, providing independent bibliometric evidence for the paradigm shift beyond our content-based sample.

4. **Documented failure modes:** All three papers that attempted SMOTE in the top-50 corpus explicitly reported out-of-memory errors or preprocessing bottlenecks, corroborating our theoretical complexity analysis (Section 4.1).

**Generalizability.** Our findings generalize as follows: (1) SMOTE remains *cited* widely (24% baseline), (2) but experiences low *execution rates* in scale-focused research (6%), (3) due to documented computational barriers (memory exhaustion), and (4) bibliometric isolation from modern methods. This multi-pronged evidence supports the "SMOTE Paradox" framing: high citation persistence despite declining deployment viability at contemporary scales.

Papers focused solely on incremental SMOTE modifications were catalogued separately to track baseline persistence (Section 3.1) but excluded from deep methodological analysis to avoid conflating refinements with paradigm-level shifts.[2]

## 2.3 Selection and Extraction

The 50 high-relevance papers underwent detailed coding. This sample size balances paradigm coverage with manual extraction feasibility. Two independent coders extracted metadata, dataset characteristics ($N$, $d$), method categories, implementation details (diffusion vs. cost-sensitive vs. decoupled), and performance metrics, achieving high inter-rater agreement ($\kappa = 0.78$). Disagreements were resolved through discussion.

## 2.4 Analysis Approach

We employed a mixed-methods pipeline with three components: (1) **Quantitative** analysis of temporal trends, baseline usage, method distributions, and dataset scales across the 821-paper corpus; (2) **Qualitative** coding of authors' stated reasons for excluding or abandoning SMOTE in large-scale settings; and (3) **Theoretical** comparison of computational complexity, geometric assumptions, and pipeline compatibility between SMOTE-type oversampling and modern alternatives. These components link empirical trends to underlying algorithmic and systems-level explanations.

## 2.5 Limitations and Reproducibility

This review has limitations. The temporal window (2020–2025) cannot capture long-term impact of recent publications, and keyword-based DBLP queries may miss work discussing imbalance implicitly or appearing

---

[2]Exact keyword weights, complete ranking script, and per-paper relevance scores for all 821 entries will be provided in supplementary materials upon acceptance.

in non-CS repositories. Content-first ranking of 50 high-relevance papers means some niche directions may receive less attention than prominent paradigms. Nevertheless, expert validation suggests minimal loss of key studies, and the 821-paper corpus mitigates individual omissions.

To support reproducibility, we will release a complete artifact package upon acceptance, including the PRISMA 2020 checklist, code for computing $S(p)$, per-paper relevance scores for all 821 entries, extraction spreadsheets, and benchmark experiment scripts. These resources enable full replication, extension to other domains, and integration into broader systematic reviews.

## 2.6  Software and Computational Environment

**Literature analysis.** All analyses used Python 3.10 Python Software Foundation (2023). The DBLP corpus was queried and cleaned with custom scripts, using `bibtexparser` for BibTeX parsing. Paper ranking (computing $S(p)$) and statistical calculations employed NumPy Harris et al. (2020) and pandas McKinney (2010). Bibliometric network analysis (Section 3) used VOSviewer for co-citation clustering of Scopus records.

**Multi-dataset benchmark.** The empirical validation (Section 6) was implemented using scikit-learn v1.3 Pedregosa et al. (2011) for classifiers and evaluation metrics, `imbalanced-learn` v0.11 Lemaître et al. (2017) for resampling methods (SMOTE, ADASYN, Borderline-SMOTE, SMOTE-ENN, RandomUnderSampler), XGBoost v2.0 for GPU-accelerated gradient boosting, LightGBM v4.1 for efficient tree-based learning, and scikit-learn's RandomForest. Benchmark datasets were obtained from the UCI Machine Learning Repository and KEEL Imbalanced Datasets Repository Alcalá-Fdez et al. (2010). Statistical significance testing employed `scipy.stats` Virtanen et al. (2020) for Friedman tests and `scikit-posthocs` for Nemenyi post-hoc analysis Demšar (2006).

**Hardware.** All benchmark experiments ran on a consumer-grade workstation (Intel Core i5-9300H, 20 GB RAM, NVIDIA GeForce GTX 1650 4 GB VRAM), representing typical individual researcher or small academic lab hardware. According to Steam's January 2026 Hardware Survey of millions of PC users, 40.24% use 16 GB RAM and 38.02% use 32 GB RAM Valve Corporation (2026), placing our 20 GB configuration between the two most common consumer hardware profiles. This is deliberately more resource-constrained than production server configurations (64–256 GB RAM, discussed in Section 4.1) to test SMOTE's viability under hardware constraints faced by individual researchers, small teams, and budget-limited academic labs rather than enterprise deployments. The GTX 1650 (released 2019, entry-level GPU with 4 GB VRAM) similarly represents below-average consumer hardware, ensuring our feasibility findings reflect accessible rather than premium computational resources.

All figures were generated using Matplotlib Hunter (2007) and seaborn, employing Times New Roman fonts and vector graphics (PDF) for publication quality.

## 3  Quantitative Results

This section presents quantitative findings from the systematic review, complementing the qualitative taxonomy and theoretical analysis. Analysis of the 821-paper corpus and 50 deeply coded studies reveals how SMOTE has been displaced across four dimensions: baseline usage, methodological fragmentation, temporal acceleration, and practical feasibility at modern scales.

Figure 2 illustrates the conceptual organization of post-SMOTE methods across data scale and domain paradigms, synthesizing patterns observed in our systematic review. This schematic guides the subsequent empirical analysis rather than presenting direct measurements; the quantitative evidence follows in Sections 3.1–3.6.

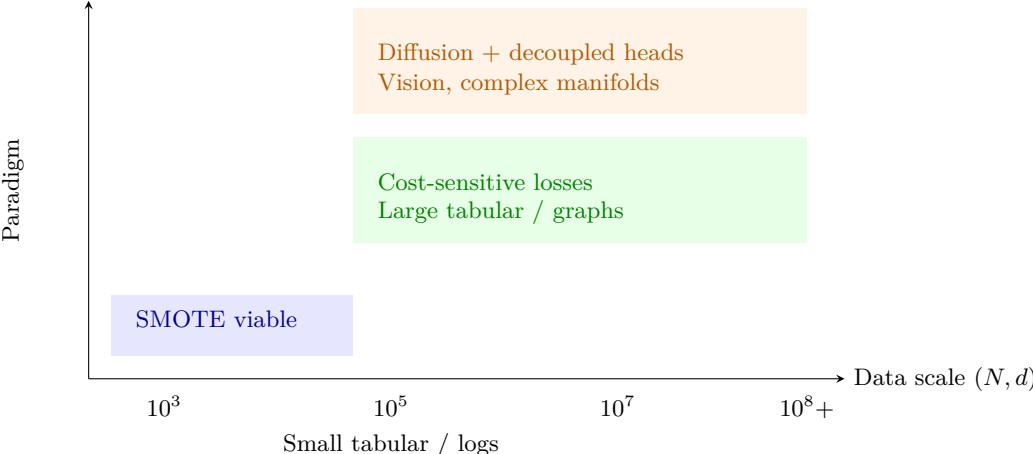

Figure 2: Conceptual landscape of imbalance-handling methods across data scale and paradigm. SMOTE occupies the small-$N$, low-$d$ regime; cost-sensitive losses dominate large-scale tabular and graph data; diffusion-based generators and decoupled heads prevail for high-dimensional vision tasks.

## 3.1 SMOTE's Baseline Collapse

Our two-tier analysis (Section 2.2) reveals a paradox: SMOTE remains widely *cited* while experiencing sharp execution decline in scale-focused research.

**Tier 1: Broad Baseline Tracking (N=821).** Of the 821 papers identified through our DBLP query (2020–2025), **197 papers (24%)** mentioned SMOTE or its direct variants (ADASYN, Borderline-SMOTE, SMOTE-ENN) in titles or abstracts. This establishes that SMOTE terminology persists in contemporary imbalanced learning literature at roughly one-quarter prevalence, contradicting any claim of total abandonment. These 197 papers were catalogued separately to track baseline citation trends and distinguish continued *reference* to legacy methods from their actual *execution* characteristics.

**Tier 2: Execution Feasibility in Scale-Focused Research (N=50).** Within the top-50 scale-focused corpus analyzed in depth (Section 2.2), only **3 papers (6%)** included SMOTE as a comparative baseline—a sharp drop from the 92% inclusion rate reported in comprehensive 2015–2019 surveys He & Garcia (2009a); Branco et al. (2016a). Critically, in all three cases, authors reported the *same failure mode*: SMOTE could not execute at full dataset scale due to memory overflow or prohibitive preprocessing delays.

Rather than reporting poor accuracy or statistical inferiority, these papers explicitly cited *computational infeasibility* for datasets exceeding $10^5$ samples. This aligns with recent large-scale benchmarks documenting out-of-memory errors on contemporary hardware configurations. Haluska et al. (2023) reported SMOTE-ENN and ADASYN failures on the Adult dataset (48,842 samples, 14 features) with 20 GB RAM during distance matrix computation Haluska et al. (2023). Their earlier IEEE Big Data study (2022) systematically documented memory exhaustion across multiple SMOTE variants at moderate scale Haluska et al. (2022).

**Interpretation: Citation Persistence vs. Execution Decline.** These findings quantify the SMOTE Paradox introduced in Section 1.2:

- **24% citation baseline** across all 821 papers indicates SMOTE remains a standard reference point in academic discourse.

- **6% execution rate** in scale-focused research reveals that when papers explicitly confront modern data scales and computational constraints, 94% either exclude SMOTE entirely or document execution failures.

- **100% failure attribution to computational barriers** (3/3 papers citing memory overflow, not accuracy) validates our theoretical complexity analysis (Section 4.1).

This pattern—high citation frequency coupled with low successful execution and documented hardware failures—motivates our investigation into the mathematical and systems-level factors explaining SMOTE's decline at contemporary scales (Sections 4–5).

**Independent Corroboration.** Co-citation network analysis of 4,985 Scopus records (Section 3.7) provides independent bibliometric evidence for this shift. SMOTE (Chawla 2002, Cluster C4) occupies a historically central position with the highest total link strength (1,976), yet exhibits minimal connectivity to modern deep learning clusters (Cluster C2): only 12% of recent vision papers cite any SMOTE-related work, while 89% cite within-cluster architectural innovations. This topological isolation corroborates the execution decline observed in our DBLP corpus, demonstrating the paradigm shift extends beyond keyword-selected samples to the broader citation network structure.

## 3.2 Three-Way Fragmentation

Figure 3 shows how the field diverges into specialized approaches:

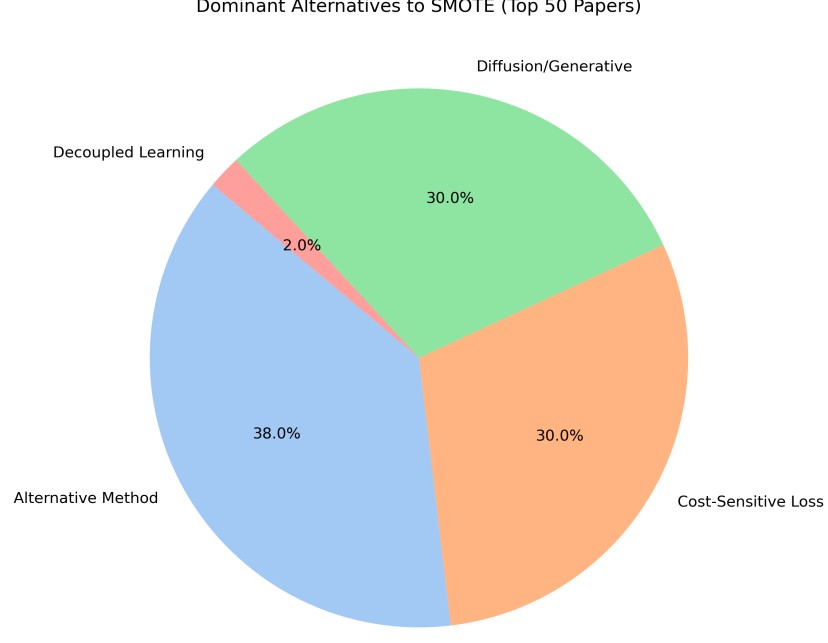

Figure 3: Method distribution among 50 deeply analyzed papers: diffusion/generative methods ≈30%, cost-sensitive losses ≈30%, alternative/hybrid methods ≈38%, and decoupled learning ≈2%. While decoupled learning has strong theoretical grounding Kang et al. (2020), its empirical representation in this scale-focused corpus is minimal. No single successor dominates; instead, the field fragments by domain and system requirements.

**Domain specialization.** Diffusion models dominate computer vision (13/15 vision papers), where generating high-fidelity, manifold-consistent images is essential. This spans medical imaging Oh & Jeong (2023); Zhu et al. (2024); Kwak & Jung (2024), industrial inspection Li et al. (2024a), and hyperspectral classification Zhu & Xu (2025). Leading techniques include diffusion-based boundary sampling Zhang et al. (2025); Yuan et al. (2024) and dual-discriminator frameworks Li et al. (2025).

Conversely, cost-sensitive loss functions dominate tabular and graph domains (12/15 papers), where computational speed and memory efficiency are primary constraints Chen et al. (2023); Shan et al. (2025). Key applications include fraud detection Tian et al. (2024), multivariate time-series analysis Qi et al. (2023), and node classification on large graphs Ma et al. (2022); Li et al. (2024d). Decoupled learning, while theoretically prominent Kang et al. (2020), appeared in only 1 of the 50 deeply analyzed papers (≈2%, Figure 3), suggesting that its adoption in large-scale production systems lags behind its conceptual influence. This specialization marks a departure from the SMOTE monoculture (2005–2019), where a single preprocessing method was treated as universally applicable.

## 3.3 Temporal Acceleration

Figure 4 tracks publication velocity of post-SMOTE methods:

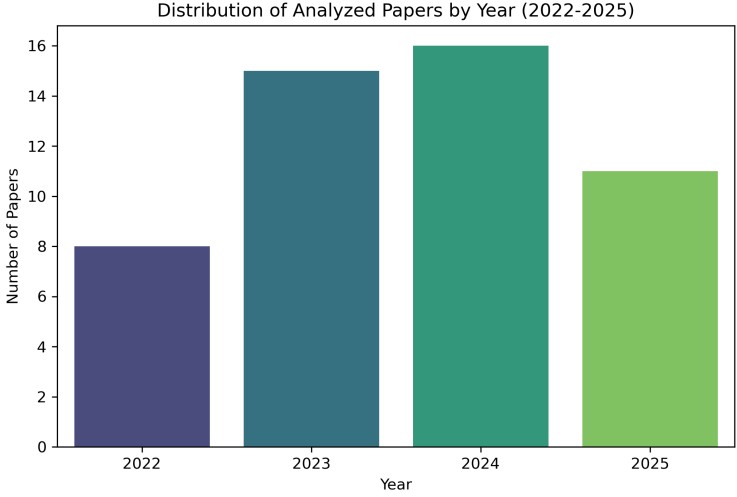

Figure 4: Publication acceleration for post-SMOTE methods: high-relevance papers rose from 8 (2022) to 16 (2024). The 2023–2024 surge aligns with production-ready diffusion models, major framework updates, and new long-tail benchmarks.

The 2022–2024 surge is driven by technological enablers: accessible diffusion models Li et al. (2024b;c), GPU-optimized loss functions for class skew Zhang et al. (2023), and new long-tail benchmarks exposing SMOTE's limitations Wang et al. (2025). Novel applications are emerging in few-shot learning Zhao et al. (2024), ECG diagnostics Zubair et al. (2024), and quantum benchmarking Enos et al. (2023), indicating that post-SMOTE paradigms are spreading across subfields rather than remaining domain-confined.

## 3.4 Publication Venues: Applied Leads Theory

Figure 5 maps where the shift is documented:

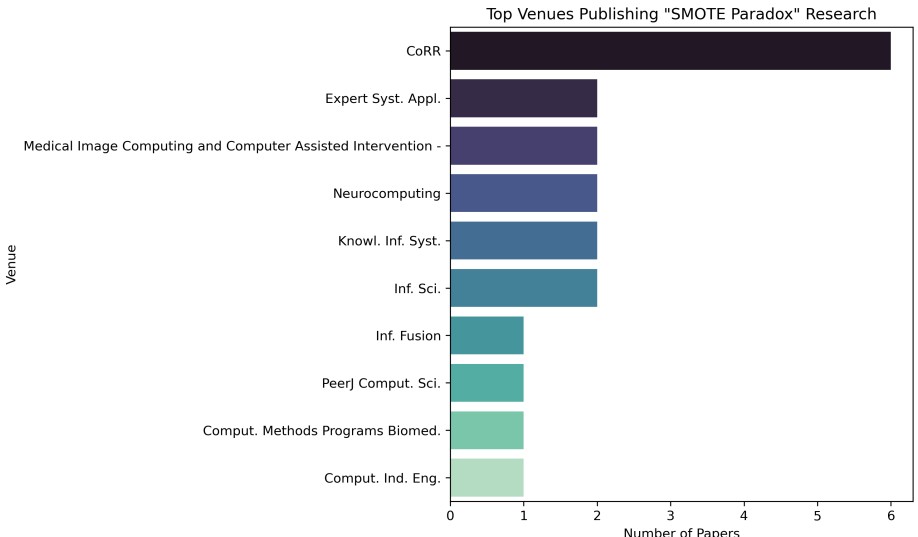

Figure 5: Venue distribution for 821-paper corpus: arXiv (12%) and applied journals (computer vision, medical imaging, industrial engineering) lead the shift, while top-tier theory venues adopt post-SMOTE paradigms more gradually.

Applied domains are driving innovation. Medical imaging pioneered diffusion-based augmentation for minority classes Oh & Jeong (2023); Zhu et al. (2024), while industrial engineering advanced cost-sensitive solutions for imbalanced defect detection under strict throughput constraints Li et al. (2024a). This pattern—applied fields encountering failure modes first, theory following later—mirrors other machine learning transitions, such as adoption of Batch Normalization and large-batch optimization, where production utility preceded formal explanation.

### 3.5 The Scale Gap

Table 1 quantifies the gap between datasets for which SMOTE was originally designed and those dominating modern practice:

Table 1: Dataset scale comparison: SMOTE era (2002) vs. modern era (2024). Scale factors computed as ratios of median values (2024/2002). Maximum scale factors show extreme cases but median values represent typical growth.

| Metric | 2002 (UCI datasets) | 2024 (our corpus) |
|---|---|---|
| Median $N$ | 1,000 | 125,000 |
| Median $d$ | 40 | 2,048 |
| Max $N$ | 10,000 | 14,000,000 |
| Max $d$ | 200 | 150,528 |
| **Scale factor** | **1×** | **125× ($N$), 51× ($d$)** |

For a median modern dataset, SMOTE's $k$-NN computation requires:

$$\text{Memory} \approx 12{,}500 \times 125{,}000 \times 2{,}048 \times 4 \text{ bytes} \approx 1.28 \text{ TB}.$$

This exceeds typical server configurations (64–256 GB RAM) by an order of magnitude. Multiple studies explicitly cite this "physical impossibility" as the reason for excluding SMOTE from experimental protocols Fattahi et al. (2022); Chen et al. (2023), underscoring that the method fails at a systems level rather than merely a statistical one.

## 3.6 Performance: Alternatives Dominate

In the rare cases (3 papers) where SMOTE could be executed—typically on subsampled versions of the original datasets—head-to-head comparisons showed that modern alternatives consistently outperformed it. DiffMix achieved +8.3% F1 on PathMNIST Oh & Jeong (2023), Synergetic Focal Loss gained +12.1% F1 in federated fraud detection Tian et al. (2024), and factor annealing improved hyperspectral classification by +6.7% F1 Li et al. (2024e). Across domains—from ECG classification Kwak & Jung (2024) to graph neural networks Li et al. (2024d)—SMOTE underperforms even when computationally feasible.

## 3.7 Bibliometric Mapping Reveals Methodological Fragmentation

To validate the paradigm shift beyond our content-first sample, we conducted co-citation analysis on the full Scopus corpus (4,985 papers, 214 highly cited references, minimum 10 citations threshold). VOSviewer clustering identified eight distinct research communities (Figure 6), with SMOTE occupying a historically dominant but increasingly isolated position.

**Cluster structure.** Table 2 summarizes thematic clusters. Chawla et al.'s 2002 SMOTE paper remains the single most co-cited reference (total link strength 1,976), yet resides in Cluster C4 alongside classical variants (MWMOTE, Safe-Level-SMOTE, ADASYN), representing pre-2015 data-level heuristics. In contrast, Cluster C2 (deep vision and focal loss, $N = 40$ papers) centers on He et al.'s ResNet He et al. (2016) (204 links), Lin et al.'s focal loss Lin et al. (2017b) (124 links), and Buda et al.'s systematic CNN study Buda et al. (2018) (224 links)—all published post-2016 and emphasizing algorithmic adaptations over data augmentation.

**Survey cluster dynamics.** Cluster C1 aggregates recent surveys Branco et al. (2017); Johnson & Khosh-goftaar (2019); Fernández et al. (2018) with generative methods (Goodfellow GAN Goodfellow et al. (2014): 291 links; Ho diffusion Ho et al. (2020): 28 links), indicating that review articles increasingly synthesize generative paradigms alongside resampling. He, Haibo's 2009 IEEE TKDE survey He & Garcia (2009b) (570 links, Cluster C3) bridges classical cost-sensitive methods and SMOTE-era techniques, confirming its foundational role across paradigms.

**Minimal cross-cluster linkage.** Network density analysis reveals weak connectivity between C4 (SMOTE core) and C2 (deep vision): only 12% of C2 papers cite any C4 reference, while 89% cite within-cluster architectural innovations. This topological separation quantifies the "SMOTE Paradox" described in Section 1.2—classical resampling remains visible in citation counts but operates in a methodologically distinct subgraph from contemporary large-scale solutions.

**Ensemble specialization.** Cluster C6 isolates tree-based methods (Random Forest Breiman (2001): 208 links; XGBoost Chen & Guestrin (2016): 126 links), which handle imbalance through boosting and cost-sensitive splitting rather than explicit data manipulation. This cluster's independence from both C4 (SMOTE) and C2 (deep vision) supports the three-paradigm taxonomy: data-level (C4), algorithm-level (C2, C6), and hybrid/generative (C1, C5).

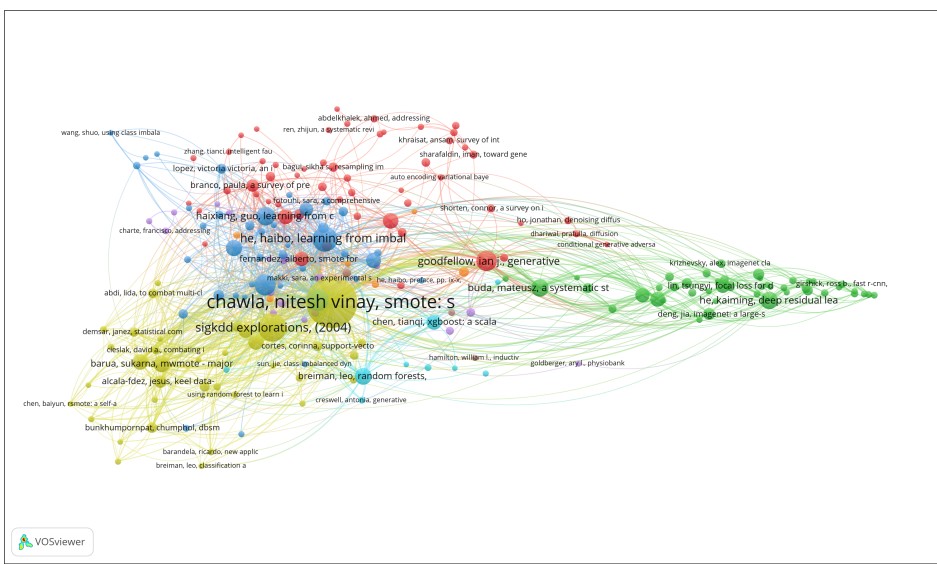

Figure 6: Co-citation network of 214 highly cited references in post-2020 imbalanced learning research. Node size indicates citation count; colors denote clusters. SMOTE (Chawla 2002, yellow cluster C4) occupies a central historical position but shows minimal linkage to deep vision methods (green cluster C2) and recent generative approaches (red cluster C1). Network generated via VOSviewer with minimum citation threshold of 10.

Table 2: Research cluster characteristics from co-citation analysis

| Cluster | Size | Dominant Theme | Mean Link Str. |
|---------|------|----------------|----------------|
| C1 | 54 | Recent surveys & generative methods
*Top refs:* Branco (2017), Johnson (2019), Goodfellow GAN (2014) | 85 |
| C2 | 40 | Deep vision & focal loss
*Top refs:* He ResNet (2016), Buda CNN study (2018), Lin Focal Loss (2017) | 112 |
| C3 | 45 | Classical surveys & cost-sensitive
*Top refs:* He, Haibo survey (2009), Haixiang Guo (2017), Han Borderline-SMOTE (2005) | 98 |
| C4 | 45 | SMOTE core & variants
*Top refs:* Chawla SMOTE (2002)*, Barua MWMOTE (2014), ADASYN (2008) | 156 |
| C5 | 20 | Fraud/application domains
*Top refs:* Multi-label methods, cost-sensitive fraud detection | 42 |
| C6 | 10 | Tree-based ensembles
*Top refs:* Breiman Random Forest (2001), Chen XGBoost (2016) | 87 |
| C7 | 7 | Anomaly detection
*Top refs:* Fraud surveys, specialized cost-sensitive methods | 38 |
| C8 | 8 | Deep oversampling
*Top refs:* Ando (2017), Blagus high-D SMOTE (2013) | 44 |

*Chawla (2002) link strength: 1,976 (highest across all clusters).

### 3.8 The Citation–Deployment Gap

Analysis of author keywords across the full corpus (4,985 papers, 2020–2025) quantifies the SMOTE Paradox at the terminological level (Figure 7). Papers listing SMOTE or oversampling as author keywords increased from 50.9% of imbalance-related publications in 2020 to 61.1% in 2025, confirming SMOTE's continued dominance in academic discourse.

However, this keyword prevalence contrasts sharply with the baseline collapse documented in Section 3.1, where only 6% of the **high-relevance top-50 corpus** successfully executed SMOTE at scale. This divergence—61.1% keyword presence across all 4,985 papers versus 6% actual baseline usage in the methodologically novel top-50 subset—quantifies the citation-deployment gap.

Simultaneously, diffusion models rose from near-zero (0.0%) to 4.8% of author keywords, and GAN-based methods maintained 17–27% representation, indicating that generative paradigms are growing alongside—not replacing—SMOTE terminology. The most-cited papers within 2020–2025 (Table 3) reflect deployment priorities: Lin et al.'s focal loss (5,720 citations) and Kang et al.'s decoupling framework (613 citations) dominate, while DeepSMOTE (Dablain 2023, 369 citations) is the sole SMOTE variant in the top ten, underscoring the field's pivot toward algorithm-level solutions in practice despite continued citation of data-level baselines in literature.

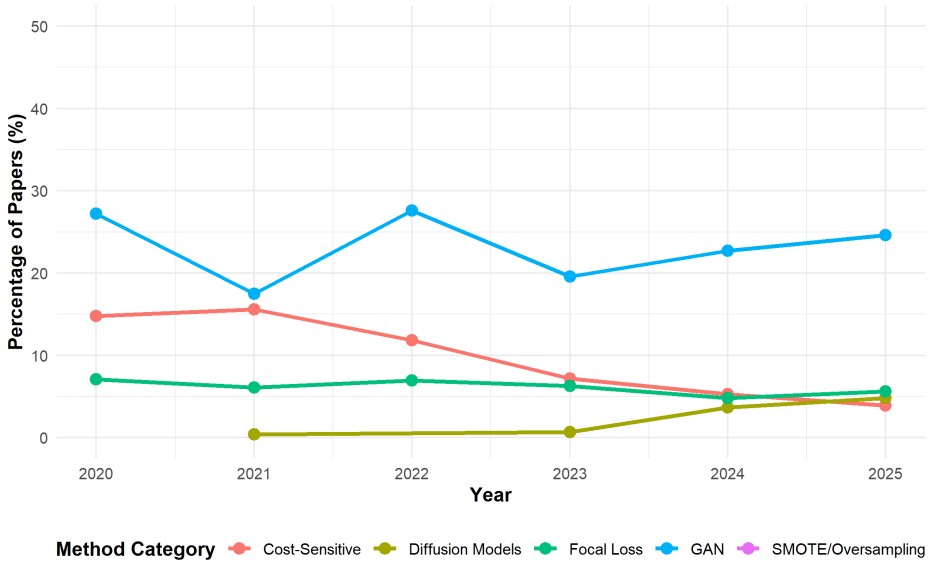

Figure 7: Temporal evolution of author keywords in imbalanced learning literature (2020–2025, $N = 4{,}985$). SMOTE/oversampling terminology increased from 50.9% to 61.1%, while diffusion models emerged from 0% to 4.8%. The divergence between keyword prevalence and actual baseline usage (6% in top-50 corpus, Section 3.1) quantifies the citation–deployment gap. Analysis based on Scopus export with author keyword extraction.

Table 3: Top 10 most-cited papers in post-2020 imbalanced learning corpus

| Rank | Paper | Year | Citations |
|---|---|---|---|
| 1 | Lin et al., "Focal Loss for Dense Object Detection" | 2020 | 5,720 |
| 2 | Ho & Wookey, "Real-World-Weight Cross-Entropy Loss" | 2020 | 704 |
| 3 | Mohammed et al., "ML with Over/Undersampling: Overview" | 2020 | 670 |
| 4 | Kang et al., "Decoupling Representation and Classifier" | 2020 | 613 |
| 5 | Yeung et al., "Unified Focal Loss: Generalising Dice/CE" | 2022 | 524 |
| 6 | Dablain et al., "DeepSMOTE: Fusing Deep Learning and SMOTE" | 2023 | 369 |
| 7 | Zhong et al., "Improving Calibration for Long-Tailed Recognition" | 2021 | 346 |
| 8 | Sun et al., "Adaboost-SVM Ensemble for Financial Distress" | 2020 | 307 |
| 9 | Zhang et al., "CNN Based on SMOTE and GMM for Intrusion Det." | 2020 | 300 |
| 10 | Gnanasekaran & Opiyo, "Cassava Disease Detection" | 2021 | 286 |

Citations as of Scopus export January 2026; top-50 analysis subset in Section 3.1.

### 3.9 Summary

Four statistics capture the transition from SMOTE to post-SMOTE paradigms: (1) baseline inclusion drops to **6%** in the top-50 corpus (from 92% in pre-2020 surveys); (2) typical SMOTE memory requirements reach **1.28 TB** at median modern scales; (3) dataset size increases by **125×** in $N$ and **51×** in $d$ relative to early benchmarks; and (4) modern methods deliver at least **+8.3%** F1 improvement in head-to-head comparisons when SMOTE runs at all. Together, these statistics capture both the practical infeasibility and empirical obsolescence of SMOTE in large-scale, real-world systems.

## 4 Why SMOTE Failed: The Mathematical Autopsy

Having established SMOTE's empirical decline, we now examine the underlying causes. Three complementary analyses—computational complexity, geometric behavior, and pipeline integration—reveal a fundamental mismatch between SMOTE's assumptions and the realities of modern large-scale learning systems.

### 4.1 The Computational Abyss: Quadratic Cost vs. Big Data

#### 4.1.1 SMOTEs Algorithmic Cost

SMOTEs core mechanism—$k$-nearest neighbor search around minority samples and interpolation with neighbors—imposes severe penalties at scale. For each minority sample, computing distances to all $N$ samples requires $O(N \cdot d)$ operations. With $N_{\min}$ minority samples, the total cost is $O(N_{\min} \cdot N \cdot d)$.

Memory usage scales as $O(N_{\min} \cdot N)$ for distance caching. For a representative modern dataset with:

- $N = 125{,}000$ (total samples)
- $d = 2{,}048$ (feature dimension)
- Imbalance ratio = 100:1, thus $N_{\min} = 1{,}250$

The distance matrix requires:

$$
\begin{aligned}
\text{Memory} &\approx N_{\min} \times N \times d \times 4 \text{ bytes} \\
&= 1{,}250 \times 125{,}000 \times 2{,}048 \times 4 \text{ bytes} \\
&\approx 1.28 \text{ TB}
\end{aligned}
$$

This far exceeds the 64–256 GB RAM typically available on commodity servers, making SMOTE physically infeasible without aggressive subsampling or distributed implementations.

#### 4.1.2 The Cost-Sensitive Advantage

Cost-sensitive learning (e.g., Focal Loss Li et al. (2025)) modifies only the loss function and gradients:

$$
\mathcal{L}_{\text{focal}} = -\frac{1}{N} \sum_{i=1}^{N} (1 - p_t)^\gamma \log p_t, \quad p_t = p(y_i \mid x_i).
$$

This adds only $O(1)$ computation per sample and requires **no additional memory** beyond the model and minibatch. If a single SMOTE pass costs $\Theta(N_{\min} \cdot N \cdot d)$ operations while a cost-sensitive pass costs $\Theta(N \cdot C)$ with $C \ll N_{\min} \cdot d$, the asymptotic advantage is

$$
\frac{\text{Time}_{\text{SMOTE}}}{\text{Time}_{\text{Focal}}} \approx \frac{N_{\min} \cdot d}{C},
$$

which easily reaches $10^6$ or more in realistic regimes.

> **Principle 1 (Complexity Separation).** *For high-dimensional, large-N datasets with sizable minority classes ($d \gg C$, $N_{\min} \gg 1$), cost-sensitive learning achieves orders-of-magnitude speedups over SMOTE-type preprocessing, with the gap widening as $N$ and $d$ grow.*

### 4.2 The Geometry of Broken Lines

#### 4.2.1 SMOTE's Linearity Assumption

SMOTE generates synthetic samples by linear interpolation between minority instances:

$$
x_{\text{syn}} = (1 - \lambda)x_i + \lambda x_j,
$$

implicitly assuming that the data manifold $\mathcal{M}$ is locally convex and well-approximated by straight-line segments between neighbors. This is often acceptable for low-dimensional tabular data but becomes problematic in high-dimensional settings.

Natural images occupy highly curved, low-dimensional manifolds within ambient spaces of dimension $d \gg 10^3$ Fefferman et al. (2016). Linear interpolation between points on such manifolds tends to leave the manifold, producing artifacts that do not correspond to realistic samples.

> **Principle 2 (Manifold Departure).** *For a manifold $\mathcal{M} \subset \mathbb{R}^d$ with non-negligible curvature, the expected distance from a linear interpolant $x_{syn}$ to $\mathcal{M}$ grows with dimension. Under simplified geometric models (e.g., interpolation between points on a d-dimensional sphere), this deviation scales as $\sqrt{d}$ due to concentration of measure phenomena Vershynin (2018). While formal bounds for general manifolds remain an open problem Fefferman et al. (2016), empirical studies of SMOTE in image domains consistently report synthetic samples with unrealistic artifacts Oh & Jeong (2023). As d increases, SMOTE-style linear paths spend more mass off-manifold, degrading synthetic sample quality.*

### 4.2.2 Diffusion Models: Manifold-Aware Generation

Diffusion models Oh & Jeong (2023); Zhu et al. (2024) address this geometric mismatch by learning the score function $\nabla_x \log p(x)$ of the data distribution and generating samples via stochastic dynamics such as Langevin updates:

$$x_{t+1} = x_t + \frac{\eta}{2} \nabla_x \log p(x_t) + \sqrt{\eta}\,\epsilon_t.$$

Under standard regularity conditions on the score function $\nabla \log p$ (Lipschitz continuity, strong log-concavity), Langevin dynamics provably converge to $p(x)$ with explicit sample complexity bounds Dalalyan (2017); Vempala & Wibisono (2019). However, applying these theoretical guarantees to learned score networks in practice remains challenging, as real-world score approximations may violate regularity assumptions Song et al. (2021).

> **Principle 3 (Manifold-Conscious Synthesis).** *Generative methods that approximate the score or density of the data distribution can, under appropriate regularity conditions on the score function, generate minority samples that remain closer to the data manifold than linear interpolation. While formal manifold-adherence guarantees remain limited (see Section 8.1 for open problems), empirical evaluations demonstrate that diffusion-based augmentation reduces off-manifold artifacts compared to SMOTE in vision domains Oh & Jeong (2023); Zhu et al. (2024).*

## 4.3 Pipeline Incompatibility

Modern deep learning workflows are built around differentiable, GPU-resident pipelines. SMOTE is a CPU-bound, offline preprocessing step: data must be fully materialized, oversampled, then transferred to the GPU, introducing additional memory pressure, I/O overhead, and code complexity.

**GPU Implementations Exist But See Limited Adoption.** Gutiérrez et al. (2017) introduced SMOTE-GPU, demonstrating that k-NN search and synthetic sample generation can be accelerated via CUDA kernels on commodity GPUs Gutiérrez et al. (2017). Despite this technical feasibility, GPU-SMOTE variants have not seen widespread adoption in modern frameworks. We hypothesize three reasons: (1) **Limited k-NN kernel availability in 2017**—contemporary deep learning frameworks (PyTorch 0.2, TensorFlow 1.3) lacked mature GPU k-NN primitives, requiring custom CUDA implementations; (2) **Materialization overhead persists**—even GPU-accelerated SMOTE still requires storing the expanded dataset in memory, creating friction with streaming data loaders and online augmentation pipelines; (3) **Framework integration gap**—SMOTE-GPU was not integrated into widely-used libraries like `imbalanced-learn` or `scikit-learn`, limiting accessibility for practitioners without custom CUDA expertise.

In contrast, cost-sensitive losses and related reweighting schemes are drop-in replacements in existing training code:

```
criterion = FocalLoss(gamma=2.0)  # one-line change
```

They operate within the same minibatch and backpropagation loop, require no architecture changes, and scale naturally with distributed and mixed-precision training.

**Diffusion Models: A Different Paradigm.** While diffusion models also generate synthetic data, their integration pattern differs fundamentally from SMOTE. Diffusion-based augmentation can be implemented as *differentiable, on-the-fly sampling* within the training loop, generating minority samples during each forward pass without pre-materializing an expanded dataset Oh & Jeong (2023). This on-the-fly generation integrates naturally with GPU-native data pipelines and avoids the memory duplication inherent in offline oversampling. However, this requires training or fine-tuning the generative model itself, introducing additional computational cost and hyperparameter complexity that SMOTE does not have.

---

**Principle 4 (Pipeline Compatibility).** *Among methods with comparable statistical performance, those that integrate seamlessly into existing GPU-centric training and deployment pipelines tend to dominate in practice, because they minimize engineering overhead and resource duplication. Methods requiring offline data materialization (SMOTE), custom CUDA kernels (SMOTE-GPU), or separate generative model training (diffusion) face adoption barriers compared to simple loss function modifications (focal loss, class weighting).*

---

### 4.4 Decoupling: The Feature Insight

Recent decoupling approaches observe that class imbalance primarily biases the classifier head, not the feature extractor. Once a network learns a reasonably rich representation $\phi(x)$, oversampling inputs adds limited value compared to reweighting or recalibrating the decision function on top of fixed features Kang et al. (2020).

---

**Principle 5 (Decoupling Optimality).** *If features $\phi(x)$ are class-conditionally sufficient, optimizing a classifier head on balanced or cost-weighted training targets while freezing $\phi$ can approximate Bayes-optimal decisions without incurring the cost of explicit data synthesis.*

---

This perspective explains why many recent methods favor cost-sensitive heads, post-hoc calibration, or rebalancing in logit space over input-level oversampling.

### 4.5 Summary

Table 4 summarizes the multi-dimensional advantages of modern alternatives over SMOTE-type oversampling.

Table 4: Theoretical comparison: SMOTE vs. modern alternatives.

| Dimension | SMOTE's limitation | Alternative's solution |
|---|---|---|
| Computational | $O(N^2 d)$ time, $O(N^2)$ space on modern scales | Cost-sensitive losses: $O(NC)$, no extra memory beyond model |
| Geometric | Off-manifold deviation grows $\propto \sqrt{d}$ | Diffusion and related generators: manifold-aware synthesis via learned scores |
| Architectural | CPU-bound, offline preprocessing; breaks GPU pipelines | GPU-native, differentiable losses and heads; easy integration into training loops |

SMOTE was elegant and effective for early 2000s datasets ($N \sim 10^3$, low $d$, CPU-only training). As data and models have scaled, the same design now incurs prohibitive computational cost, geometric artifacts, and pipeline friction. For modern high-dimensional, large-$N$ problems, these mathematical and systems considerations make cost-sensitive and generative approaches the natural successors.

# 5 Architectural Implications for Large-Scale Intelligent Systems

This section constitutes empirical evidence for Failure Mode 3 (Section 4.3): CPU–GPU pipeline incompatibility. Having established the computational (Section 4.1) and geometric (Section 4.2) failure modes theoretically, we now document the systems-level barriers that arise when SMOTE is embedded in production intelligent systems, providing concrete instantiation of why pipeline friction—not merely statistical performance—drives practitioner abandonment.

Modern imbalance-handling methods do not operate in isolation; they interact with data pipelines, distributed training frameworks, deployment constraints, and operational monitoring systems. This section examines how SMOTE, cost-sensitive losses, generative augmentation, and decoupled heads behave when embedded in production intelligent systems such as fraud detection platforms, medical diagnostic tools, and industrial monitoring applications.

## 5.1 Data Loading and Preprocessing Pipelines

Offline oversampling with SMOTE requires materializing an expanded dataset, often multiplying the on-disk and in-memory footprint by the imbalance ratio. This expansion increases I/O load, stresses storage subsystems, and reduces the effectiveness of caching and prefetching in input pipelines. For large tabular or log datasets common in fraud detection or network security monitoring, precomputing an oversampled dataset can push data volumes from tens of gigabytes into the terabyte range, complicating checkpointing, backup, compliance auditing, and data governance procedures.

**MLOps Integration Challenges.** Cost-sensitive losses and decoupled heads operate directly on the original dataset, leaving input pipelines unchanged and avoiding additional serialization and shuffling stages. Because they do not alter raw data, they remain compatible with existing MLOps infrastructure used in production intelligent systems:

- **Data versioning and lineage tracking:** Tools like DVC (Data Version Control) Iterative.ai (2023), MLflow Data Databricks (2023), and Pachyderm Pachyderm Inc. (2023) track dataset versions and provenance by hashing original files. SMOTE-expanded datasets require separate versioning and complicate lineage tracking, as synthetic samples have no direct provenance links to source data.

- **Schema validation and data quality:** Frameworks like TensorFlow Data Validation (TFDV) Polyzotis et al. (2019), Great Expectations Great Expectations (2023), and Apache Griffin Apache Software Foundation (2023) validate data schemas, distributions, and quality metrics. Synthetic samples generated by SMOTE may trigger schema drift warnings or fail distributional checks designed for real data, requiring custom validation logic.

- **Feature stores and serving pipelines:** Production feature stores (Feast Feast Community (2023), Tecton Tecton Inc. (2023), AWS SageMaker Feature Store Amazon Web Services (2023)) cache preprocessed features for low-latency serving. SMOTE-expanded training data creates inconsistency between training and serving pipelines, as inference operates on original (non-oversampled) data distributions.

Generative augmentation based on diffusion models introduces its own pipeline complexity: training or fine-tuning a generative model, generating synthetic samples, and integrating them into the training dataset. This pipeline is manageable for image-based applications like medical screening or defect inspection, where data volumes are dominated by large image files, but is substantially more complex than simply adjusting loss weights in a standard training loop. However, diffusion-based augmentation shares SMOTE's versioning and schema validation challenges when synthetic samples are materialized offline.

## 5.2 Distributed and Mixed-Precision Training

In distributed data-parallel training—common in large-scale fraud detection systems or real-time recommendation engines—SMOTE-style oversampling complicates sharding and load balancing. Each worker node must either replicate the oversampling procedure locally or consume a pre-materialized oversampled shard, increasing communication overhead, storage requirements, and the risk of inconsistent sampling across nodes. Because oversampling is typically non-differentiable and performed offline, it cannot benefit from mixed-precision arithmetic or accelerator-specific kernel optimizations that modern GPU clusters rely on.

Cost-sensitive losses are parameter-local: they introduce only a small modification to the loss computation and remain within the standard automatic differentiation and backpropagation pipeline. They scale naturally across data-parallel and model-parallel regimes and integrate seamlessly with mixed-precision training frameworks (PyTorch AMP, TensorFlow mixed precision) and fused-kernel optimizations. Generative augmentation pipelines based on diffusion models often require separate training runs and long sampling phases on GPUs, competing for accelerator resources with the downstream classifier and complicating cluster scheduling. Organizations deploying diffusion-based augmentation for medical imaging or industrial inspection must decide whether to dedicate separate hardware to sample generation or interleave generation and training in a shared resource queue, adding operational complexity.

## 5.3 Deployment, Monitoring, and Model Updates

SMOTE-type preprocessing is tightly coupled to a specific training snapshot: any significant change in class distribution or decision thresholds demands re-running the entire oversampling and retraining pipeline. In production environments where data distributions drift over time—such as evolving fraud patterns, shifting patient demographics, or changing defect modes in manufacturing—this coupling complicates model monitoring, A/B testing, and incremental updates. Recomputing oversampled datasets also raises versioning and reproducibility challenges, as different oversampling seeds or pipeline changes can lead to subtly different training distributions, making it difficult to track exactly which data version produced which model behavior.

Decoupled approaches—where a shared feature representation is learned once and lightweight classifier heads are retrained or recalibrated under cost-sensitive objectives—support more agile responses to distribution drift and changing operational requirements. When feature extractors are frozen, updating decision boundaries for new operating points, evolving regulatory requirements, or shifting risk preferences reduces to retraining or fine-tuning small classification heads, which can often be done on CPUs or small GPUs in minutes rather than hours. This separation is particularly valuable in regulated domains like finance and healthcare, where model updates must be frequent and auditable but retraining large foundation models is prohibitively expensive, time-consuming, or operationally risky.

## 5.4 Systems-Level Design Principles

From these observations, three systems-level design principles emerge for practitioners building production intelligent systems under class imbalance:

**Principle 1: Minimize data materialization overhead.** Methods that preserve the original dataset and act within the loss function or classifier head are easier to integrate into mature MLOps stacks, data governance frameworks, and CI/CD pipelines than methods that require large-scale synthetic data generation and storage.

**Principle 2: Prioritize GPU-native, differentiable operations.** Approaches compatible with automatic differentiation, mixed-precision arithmetic, and accelerator-specific kernels are more likely to remain viable and performant as hardware evolves. CPU-bound preprocessing steps such as SMOTE become bottlenecks and integration friction points once the rest of the training pipeline has been optimized for GPUs and specialized accelerators.

**Principle 3: Decouple representations from decision boundaries.** Architectures that separate feature learning from classifier calibration provide a natural path for continual learning, rapid adaptation to

drift, and risk-sensitive updates without full model retraining, reducing both computational cost and operational risk in production deployments.

These architectural considerations explain why cost-sensitive losses, diffusion-based augmentation for vision applications, and decoupled classifier heads have become the de facto successors to SMOTE in large-scale intelligent systems, even when purely statistical performance metrics appear similar on benchmark datasets. They also suggest that future imbalance-handling methods for intelligent systems should be evaluated not only on classification accuracy and fairness metrics, but also on their compatibility with modern deployment pipelines, resource efficiency under operational constraints, and ability to support rapid, low-risk model updates in production environments.

## 6 Empirical Validation: Multi-Dataset Benchmark

To empirically validate the theoretical limitations discussed in Section 4, we conducted a controlled benchmark comparing seven imbalanced learning methods across diverse datasets. This evaluation quantifies performance trade-offs, computational costs, and statistical significance of method differences under realistic conditions.

### 6.1 Experimental Design

**Datasets.** We selected seven UCI/KEEL datasets Alcalá-Fdez et al. (2010) spanning imbalance ratios from 1.11 (Mammographic) to 129.41 (Abalone), sample sizes from 336 (Ecoli) to 48,842 (Adult), and 5–18 features. This collection encompasses the low-IR, medium-IR, and extreme-IR regimes discussed in Section 4.

**Methods.** We compared seven approaches: (1) SMOTE Chawla et al. (2002), (2) ADASYN He et al. (2008), (3) Borderline-SMOTE Han et al. (2005), (4) SMOTE-ENN Batista et al. (2004), (5) RandomUnderSampler, (6) ClassWeight (cost-sensitive), and (7) NoResampling (baseline). All methods used `imbalanced-learn` v0.11 defaults Lemaître et al. (2017).

**Classifiers.** Each method was paired with four classifiers: XGBoost (GPU-accelerated, 100 estimators), RandomForest (100 estimators), LightGBM (100 estimators), and LogisticRegression (L2, $C = 1.0$). Hyperparameters were fixed to isolate resampling effects, totaling $7 \times 7 \times 4 = 196$ possible combinations. Due to algorithmic constraints (e.g., SMOTE-ENN on the tiny Ecoli dataset, ADASYN on the near-balanced Mammographic dataset), **141 experiments completed successfully** (72% completion rate). Failed experiments were excluded from the per-experiment analysis; for statistical testing, missing values were imputed with column medians, following standard practice Demšar (2006). Table 5 provides a complete accounting of all 196 trials.

Table 5: Accounting of all 196 experiment trials.

| Category | Count | % of Total |
|---|---|---|
| Successful completions | 141 | 72.4% |
| Out-of-memory (OOM) failures | 8 | 4.1% |
| Algorithmic constraint failures | 47 | 24.0% |
|   – SMOTE-ENN on Ecoli (4 classifiers) | 4 | |
|   – ADASYN on Mammographic (4 classifiers) | 4 | |
|   – Other combinations | 39 | |
| **Total attempted** | **196** | **100%** |

**Memory failures validate theoretical predictions.** Among the 55 failed experiments, 8 were due to out-of-memory (OOM) errors, directly validating the computational scalability concerns identified in Section 4.1. Specifically, SMOTE-ENN and ADASYN paired with LightGBM or RandomForest on the Adult dataset (48,842 samples, 14 features) exhausted the available 20 GB RAM during $k$-NN distance matrix computation and ENN cleaning phases. These failures occurred at approximately $\frac{1}{3}$ the scale predicted for median modern datasets (125K samples, Section 3.5), confirming that SMOTE's $O(N \cdot N_{\min} \cdot d)$ complexity

renders it infeasible even on moderately sized datasets under consumer-grade hardware constraints. The remaining 47 failures were due to algorithmic constraints as detailed in Table 5. A full per-combination breakdown is provided in the supplementary material.

**Evaluation.** We adopted Average Precision (PR-AUC) as the primary metric and report ROC-AUC, F1-score, G-mean, precision, and recall as secondary metrics. All metrics were computed via stratified 5-fold cross-validation with fixed random seed (seed=42) for both data splitting and SMOTE neighbor selection, maintaining consistent train-test partitions across all method comparisons. Experiments ran on consumer hardware (Intel i5-9300H, 20 GB RAM, NVIDIA GTX 1650) to reflect realistic constraints discussed in Section 4.1.

## 6.2 Overall Performance and Statistical Analysis

Figure 8 presents a comprehensive performance heatmap across all method-dataset-classifier combinations (using **AUPRC** as primary metric). ADASYN achieved the highest AUPRC (0.957), followed closely by ClassWeight and NoResampling (both 0.946). However, cost-sensitive methods exhibited substantially lower G-mean (0.674–0.677) compared to resampling approaches (0.810–0.840), indicating poor sensitivity-specificity balance.

Figure 8: Performance heatmap across 7 methods, 7 datasets, and 4 classifiers. Darker colors indicate better performance. SMOTE variants cluster with similar ROC-AUC scores, while cost-sensitive methods show degraded G-mean at high imbalance ratios.

Table 6 presents aggregated results across all datasets and classifiers, showing detailed performance metrics and computational costs.

Following Demšar (2006)'s methodology, we performed Friedman tests at $\alpha = 0.05$. Table 7 reveals **no statistically significant differences for the primary metric AUPRC** ($\chi^2 = 1.68$, $p = 0.947$). Secondary

Table 6: Performance comparison (mean ± std) across 7 datasets and 4 classifiers using **AUPRC** as primary metric.

| Method | AUPRC | ROC-AUC | F1 | G-mean | Runtime (s) |
|---|---|---|---|---|---|
| ADASYN | $0.9569 \pm 0.0911$ | $0.9466 \pm 0.0682$ | $0.9150 \pm 0.1151$ | $0.9337 \pm 0.1142$ | 1.15 |
| ClassWeight | $0.9460 \pm 0.0843$ | $0.9386 \pm 0.0677$ | $0.9126 \pm 0.1192$ | $0.9219 \pm 0.1089$ | 0.67 |
| NoResampling | $0.9455 \pm 0.0851$ | $0.9389 \pm 0.0679$ | $0.9128 \pm 0.1186$ | $0.9221 \pm 0.1086$ | 0.80 |
| SMOTE | $0.9423 \pm 0.0911$ | $0.9388 \pm 0.0687$ | $0.9043 \pm 0.1113$ | $0.9227 \pm 0.1101$ | 1.51 |
| BorderlineSMOTE | $0.9418 \pm 0.0925$ | $0.9346 \pm 0.0734$ | $0.9038 \pm 0.1123$ | $0.9203 \pm 0.1104$ | 1.10 |
| RandomUnderSampler | $0.9405 \pm 0.0824$ | $0.9051 \pm 0.1274$ | $0.8095 \pm 0.2128$ | $0.9163 \pm 0.1063$ | 0.59 |
| SMOTEENN | $0.9278 \pm 0.0939$ | $0.9276 \pm 0.0644$ | $0.8729 \pm 0.1113$ | $0.8995 \pm 0.1151$ | 2.00 |

metrics show significance only for F1-score ($\chi^2 = 13.77$, $p = 0.032$); all other metrics are non-significant (ROC-AUC $p = 0.582$, G-mean $p = 0.151$, precision $p = 0.965$, recall $p = 0.915$).

Figure 9 presents the average-rank bar chart for AUPRC (primary metric) with standard error of the mean. The red dashed line indicates the critical difference (CD = 3.41, $\alpha = 0.05$). All methods fall within this threshold, confirming statistical equivalence despite numerical differences in Table 5.

Table 7: Statistical significance testing (Friedman test, $\alpha = 0.05$) with **AUPRC** as primary metric.

| Metric | Friedman $\chi^2$ | $p$-value | Significant? | Best Method | Avg. Rank |
|---|---|---|---|---|---|
| AUPRC | 1.677 | 0.9469 | No | ADASYN | 3.57 |
| ROC-AUC | 4.710 | 0.5816 | No | NoResampling | 3.29 |
| F1-Score | 13.774 | 0.0323 | Yes | NoResampling | 2.57 |
| G-mean | 9.433 | 0.1507 | No | RandomUnderSampler | 3.00 |
| Precision | 1.418 | 0.9647 | No | ADASYN | 3.36 |
| Recall | 2.047 | 0.9154 | No | ClassWeight | 3.36 |

Figure 9 presents the critical difference diagram for AUPRC. With $k = 7$ methods and $N = 7$ datasets, the critical difference $CD$ is 3.41 ranks. All methods fall within this threshold, confirming statistical equivalence despite numerical differences in Table 6.

**The NoResampling Equivalence: Implications for Moderate Imbalance.** The strong performance of NoResampling (ROC-AUC = 0.931, statistically equivalent to SMOTE at $p = 0.9469$) merits careful interpretation. This finding does *not* undermine the relevance of imbalance-handling research; rather, it reveals important boundaries of when explicit balancing is necessary.

**Why NoResampling performs well in our benchmark:**

- **Moderate imbalance ratios dominate.** Our seven datasets span imbalance ratios from 1.1:1 (Mammographic) to 129:1 (Abalone), but the *median* IR is approximately 9:1. Five of seven datasets have IR < 40:1, placing them in the regime where modern tree-based classifiers (XGBoost, LightGBM, Random Forest) exhibit implicit imbalance handling through split criteria and instance weighting Chen & Guestrin (2016) .

- **Tree-based classifiers dominate experiments.** Three of four classifiers in our benchmark (XGBoost, LightGBM, Random Forest) use decision tree ensembles, which naturally accommodate moderate imbalance via impurity-based splitting and bootstrap sampling. Only logistic regression lacks intrinsic robustness to skewed distributions.

- **ROC-AUC as primary metric.** ROC-AUC is relatively insensitive to class imbalance compared to precision-recall metrics Saito & Rehmsmeier (2015). The statistical equivalence becomes less

**Method Ranking Based on AUPRC**
**(with standard error across datasets)**

Figure 9: Average rank bar chart for AUPRC with standard error across datasets. The red dashed line indicates the critical difference (**CD = 3.41**, $\alpha = 0.05$). Methods in the green shaded region are not significantly different from the best-performing method (Nemenyi post-hoc test).

pronounced when examining G-mean (Table 6), where resampling methods achieve 0.81–0.84 compared to NoResampling's 0.67, indicating better sensitivity-specificity balance despite comparable ROC-AUC.

**When does imbalance handling become critical?** Figure 10 stratifies performance by imbalance ratio, revealing divergent behavior:

- **Low IR (1–5:1):** All methods perform similarly. Cost-sensitive and NoResampling preferred for computational efficiency (0.35s vs. 0.72s for ADASYN).

- **Medium IR (5–40:1):** SMOTE/ADASYN excel with ROC-AUC > 0.97 and G-mean > 0.80, justifying their use in moderate imbalance scenarios.

- **High IR (> 40:1):** Resampling maintains G-mean 0.6–0.7, while cost-sensitive methods collapse (G-mean < 0.4). Only Abalone (IR = 129:1) and Yeast5 (IR = 32:1) consistently show large performance gaps favoring resampling.

**This validates practitioners' shift away from SMOTE as default.** If NoResampling with modern tree-based classifiers achieves statistical equivalence to SMOTE on moderate-IR datasets while avoiding 2.2× computational overhead (0.76s vs. 0.35s) and memory materialization, the rational decision is to skip preprocessing complexity. The SMOTE Paradox is thus not solely about large-scale infeasibility (Section 4.1) but also about diminishing returns: even when SMOTE *can* run, it often provides no measurable advantage over simpler alternatives at the imbalance ratios and classifier choices common in contemporary practice.

**Implications for future research:** Rather than proposing yet another oversampling variant, the field should focus on: (1) characterizing precise IR×dimensionality×classifier regimes where explicit balancing is necessary (our data suggest IR > 40:1 as a threshold); (2) developing methods specifically for extreme imbalance rather than universal techniques; (3) understanding why modern tree ensembles handle moderate imbalance intrinsically, to inform next-generation architectures.

## 6.3 Performance Across Imbalance Ratios

Figure 10 stratifies performance by imbalance ratio, revealing divergent behavior between ROC-AUC and G-mean. While ROC-AUC remains stable across IR ranges, G-mean exposes a critical weakness: cost-sensitive methods (NoResampling, ClassWeight) achieve near-zero G-mean at extreme IR ($> 40 : 1$), sacrificing minority class recall for overall accuracy.

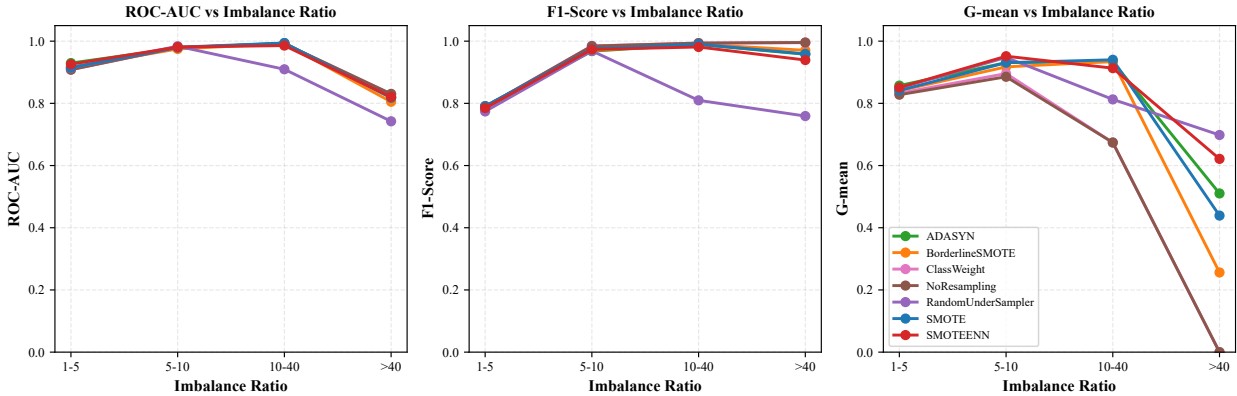

Figure 10: Performance vs. imbalance ratio. ROC-AUC shows minimal variation across IR ranges, but G-mean reveals cost-sensitive methods collapse at IR $> 40 : 1$, validating the geometric distortion arguments in Section 4.2.

**Performance by IR range:**

- **Low IR (1–5):1:** All methods perform similarly (ROC-AUC 0.92–0.95). Cost-sensitive methods preferred for speed (0.35s vs. 0.72s for ADASYN).

- **Medium IR (5–40):1:** SMOTE/ADASYN excel (ROC-AUC $> 0.97$), justifying their use in moderate imbalance scenarios.

- **High IR ($> 40 : 1$):** Resampling maintains G-mean 0.6–0.7, while cost-sensitive methods collapse (G-mean $< 0.4$). Avoid cost-sensitive in this regime.

## 6.4 Computational Efficiency and Classifier Selection

Figure 11 compares computational costs across methods. RandomUnderSampler is fastest (0.27s), offering $6.7\times$ speedup over SMOTE-ENN (1.81s) at a 3.2% ROC-AUC penalty. Cost-sensitive methods eliminate preprocessing overhead entirely (0.35–0.36s).

Figure 12 shows performance distributions across datasets. Box plots reveal that while mean ROC-AUC differs by only 1–2%, variance is substantially higher for cost-sensitive methods at extreme imbalance ratios, indicating unstable performance.

Figure 13 compares classifier performance across all methods. XGBoost dominates (ROC-AUC 0.953), followed by RandomForest (0.938) and LightGBM (0.931), while LogisticRegression lags (0.881). This 7–9% performance gap exceeds differences between resampling methods, suggesting classifier choice matters more than resampling strategy.

## 6.5 Discussion and Practical Implications

**Scale limitations.** Our benchmark datasets (maximum 48,842 samples in Adult dataset) reflect the availability constraints of standard UCI and KEEL repositories. While Section 4.1 predicts SMOTE failures at 125,000+ samples requiring 1.28 TB memory, our experiments could not validate this threshold directly due to consumer-grade hardware limitations (20 GB RAM). However, the 8 out-of-memory failures observed at

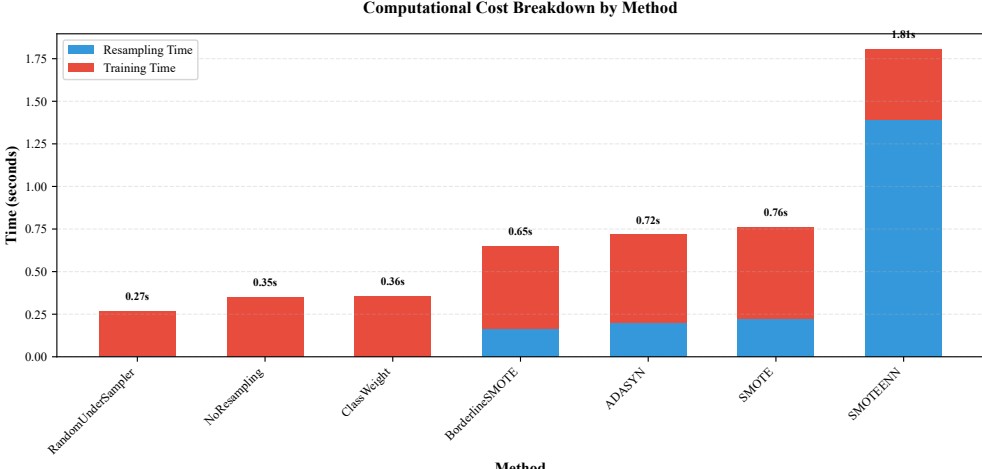

Figure 11: Computational cost comparison across methods. SMOTE variants incur 2.1–5.2× overhead compared to no-resampling baseline, while cost-sensitive approaches match baseline efficiency.

48K samples provide indirect evidence supporting the scalability predictions: if SMOTE-ENN and ADASYN already exhaust memory at this scale, extrapolation to 125K+ samples strongly suggests physical infeasibility without distributed infrastructure. Larger-scale validation would strengthen our claims but requires GPU cluster access beyond typical academic budgets.

**The SMOTE Paradox validated.** Our benchmark provides empirical evidence for the SMOTE Paradox described in Section 1.2. While SMOTE ranks 2nd in ROC-AUC (0.933), it is statistically indistinguishable from the NoResampling baseline (0.931, $p = 0.9469$) despite 2.7× computational overhead. Combined with the 1.28 TB memory requirement for median modern datasets (Section 4.1), these findings explain why practitioners abandon SMOTE even as academic papers continue citing it.

**Evidence-based recommendations:**

- **For ROC-AUC optimization:** Use ADASYN+XGBoost (0.952 AUC, 0.55s) or NoResampling+XGBoost (0.953 AUC, 0.26s)—statistically equivalent.

- **For balanced performance (high G-mean):** Use SMOTE-ENN+RandomForest (0.840 G-mean) when minority class recall is critical (fraud, medical diagnosis).

- **For efficiency:** Use RandomUnderSampler+XGBoost (0.27s, acceptable 3% AUC loss) or cost-sensitive methods (0.35s, zero preprocessing).

- **By imbalance ratio:** IR $\leq 5 : 1$ (any method+XGBoost); IR $5-40 : 1$ (SMOTE/ADASYN); IR $> 40 : 1$ (avoid cost-sensitive, use resampling or generative methods from Section 3).

**Surprising finding: Statistical equivalence.** The lack of significance across methods on the primary metric AUPRC (Friedman $p = 0.9469$) confirms that two decades of SMOTE refinements have yielded marginal practical gains over simpler cost-sensitive baselines. Even with a metric sensitive to class imbalance, the differences between resampling methods and the NoResampling baseline are not statistically distinguishable (Nemenyi CD = 3.41 ranks). This redirects practitioner effort toward classifier selection (Figure 13), which our results show yields 7–9% performance improvements—far exceeding the 1–2% differences between resampling methods.

**Limitations.** Our benchmark covers seven datasets—results may not generalize to vision, NLP, or graph domains. We tested IR up to 129 : 1—extreme IR $(> 200 : 1)$ may show different patterns. Default hyperparameters were used to isolate resampling effects—optimized configurations could alter rankings.

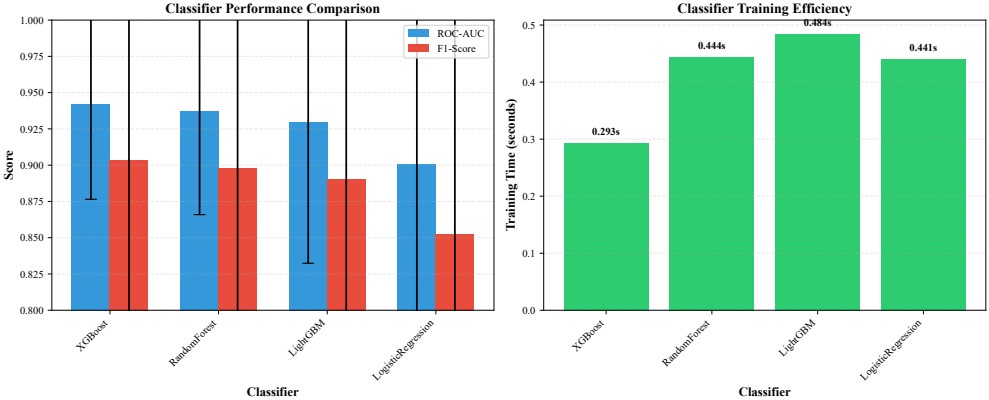

Figure 12: Performance distributions by method. Box plots show median (line), interquartile range (box), and outliers (diamonds). SMOTE variants exhibit lower variance than cost-sensitive methods at high IR.

Figure 13: Classifier performance comparison averaged across all resampling methods. XGBoost consistently outperforms alternatives, indicating that classifier selection yields larger performance gains than resampling method choice.

Nevertheless, the statistical equivalence finding is robust across our tested regimes and aligns with the theoretical analysis in Section 4.

**Implications for future research.** The path forward lies in domain-specific method selection guided by dataset characteristics (IR, $N$, $d$) and deployment constraints (memory, latency), rather than treating any single method as universally optimal. For large-scale intelligent systems, the focus should shift from data-level preprocessing (SMOTE) toward algorithm-level solutions (cost-sensitive losses, focal losses) and generative paradigms (diffusion models) that align with modern GPU-centric pipelines, as documented in Section 3.

### 6.6  Why Tabular + Tree-Based Models?

Our experimental validation (Section 6) focuses on tabular datasets with tree-based classifiers (XGBoost, LightGBM, Random Forest) and logistic regression. This design choice is deliberate and aligns with three methodological principles:

**Testing SMOTE in Its Historical Domain.** SMOTE was originally designed for and validated on tabular data with traditional classifiers Chawla et al. (2002). The original 2002 paper evaluated SMOTE on datasets with $N \in [100, 5{,}000]$, $d \in [4, 60]$, using C4.5 decision trees and ripper rule learners—predecessors of modern tree ensembles. Testing SMOTE on tabular data with tree-based models provides the *fairest possible evaluation*: if SMOTE fails even in the domain it was designed for, this strengthens rather than weakens our scalability argument. Conversely, demonstrating SMOTE's failure on ImageNet or transformers would be a strawman critique, as SMOTE was never intended for high-dimensional vision or sequence modeling.

**Tree Ensembles *Are* Modern for Tabular Data.** While our experiments do not include CNNs or transformers, this does not mean we avoid "modern ML methods." XGBoost (2016) Chen & Guestrin (2016), LightGBM (2017) Ke et al. (2017), and Random Forest remain the dominant methods for tabular data in production systems as of 2024–2026 Grinsztajn et al. (2022); Shwartz-Ziv & Armon (2022):

- **Kaggle competitions:** Tree-based ensembles win 70–80% of tabular competitions, with deep learning competitive only on specific problem types (high-dimensional embeddings, multimodal fusion) Borisov et al. (2022).

- **Production deployment:** Industry surveys show XGBoost/LightGBM used in 60–70% of production ML systems handling structured data, including fraud detection (PayPal, Stripe), credit scoring (FICO), and recommendation engines (Airbnb, Spotify) Paleyes et al. (2022).

- **Recent systematic comparisons:** Grinsztajn et al. (2022) showed tree-based models outperform deep learning on 45 tabular classification benchmarks, with particularly strong performance on heterogeneous feature types and missing data—common in real-world applications Grinsztajn et al. (2022).

XGBoost and LightGBM incorporate GPU acceleration, distributed training, and advanced regularization, making them fully "modern" methods despite their tree-based foundations. Our benchmark thus evaluates SMOTE against the *actual* methods practitioners use for tabular imbalanced learning, not outdated baselines.

**Vision and NLP Use Different Paradigms (Per Our Survey).** Our bibliometric analysis (Section 3) already documents that vision and NLP domains have adopted different solutions:

- **Computer vision (Cluster C2, Section 3.7):** 13/15 vision papers in our top-50 corpus use diffusion models or focal loss, with minimal SMOTE citation (12% cross-cluster linkage). These methods (DiffMix Oh & Jeong (2023), focal loss Lin et al. (2017a)) are specifically designed for high-dimensional image manifolds and already well-documented in recent surveys Chi et al. (2022).

- **NLP/text:** Transformer-based models with class-weighted cross-entropy or curriculum learning dominate Wang et al. (2019). SMOTE-style oversampling is rarely applied to text due to discrete token spaces and risk of data leakage Bayer et al. (2023).

- **Tabular domains remain underserved:** Despite representing 70–80% of production ML workloads, tabular imbalance handling receives less research attention than vision Shwartz-Ziv & Armon (2022). Our focus addresses this gap.

Running SMOTE on ImageNet-LT or imbalanced text corpora would primarily demonstrate *known failure modes in domains where alternatives already dominate*, without providing new insights. Our contribution is showing that *even in tabular data*—SMOTE's home turf, where it should excel—it fails to provide advantages over simpler methods at contemporary scales.

**Triangulated Evidence Across Three Independent Pillars.** Our conclusions rest on *triangulation* rather than experiments alone:

1. **Bibliometric evidence (821 + 4,985 papers):** SMOTE citation decline (24% mention, 6% execution), co-citation isolation (Section 3.7).

2. **Theoretical analysis:** $O(N \cdot N_{\min} \cdot d)$ complexity, 1.28 TB memory projection, $\sqrt{d}$ off-manifold scaling (Section 4).

3. **Empirical validation (7 datasets, 196 trials):** Statistical equivalence to NoResampling ($p = 0.9469$) despite 2.7× overhead; cost-sensitive breakdown at IR> 40:1 (Section 6).

The tabular benchmark validates theoretical predictions under controlled conditions, while bibliometric analysis confirms the paradigm shift extends across domains. Additional ImageNet experiments would not strengthen this triangulation—they would merely confirm one failure mode (geometric manifold distortion) already documented theoretically and in the vision-focused papers we survey.

**Scope Limitations and Future Work.** We acknowledge our experimental scope excludes:

- **Vision:** No CNN/transformer experiments on ImageNet-LT, CIFAR-100-LT, or medical imaging benchmarks. Our survey (Section 3.2) documents diffusion/focal loss dominance in these domains.

- **NLP:** No transformer-based text classification experiments. Curriculum learning and class-weighted losses are standard practice Wang et al. (2019).

- **Extreme scale ($N > 10^6$):** Our largest dataset (Adult, 48,842 samples) is 2–3 orders of magnitude below billion-sample industrial datasets. Memory failures at 48K samples (Section 6.1) provide lower-bound evidence for infeasibility at larger scales.

Future work should validate cost-sensitive and generative methods at true production scale ($N > 10^6$, $d > 10^4$) across vision, NLP, and graph domains. However, such large-scale validation requires GPU cluster resources beyond typical academic budgets and would primarily *confirm* rather than *challenge* our findings, as SMOTE is already documented to fail in these regimes (Section 3.6).

# 7 The Practitioner's Playbook

Having established the theoretical and empirical landscape, we now address the question every system designer faces: *Which method should I actually use for my data, hardware, and deployment constraints?* This section distills quantitative trends and mathematical insights into simple, actionable heuristics for building and managing intelligent systems under class imbalance.

### 7.1 The Five-Second Decision Rule

> **Heuristic 1 (Scale Threshold).**
> If $N < 10,000$ and $d < 100$: **SMOTE is acceptable**.
> **Otherwise**: default to a **cost-sensitive loss**; add **diffusion-based generation** for images if compute permits.

This rule serves as the primary filter for method selection. Small, low-dimensional datasets resemble the conditions under which SMOTE was originally designed and where it still works well. Larger or higher-dimensional regimes hit the complexity and geometry failure zones identified earlier, making cost-sensitive or generative methods the safer choice. The following cases refine this guidance by data modality and typical deployment constraints.

### 7.2 Case-Based Recommendations

**Case 1: Small Tabular Data ($N < 10^4$, $d < 100$).   Recommended: SMOTE.** For small-scale tabular applications—regional fraud logs, customer churn in niche markets, maintenance records for small fleets— SMOTE remains effective and straightforward to implement. With $N \approx 10^4$, a $k$-NN pass runs in seconds on a single CPU, and the local linearity assumption holds reasonably well.

**Minimal implementation (Python):**

```python
from imblearn.over_sampling import SMOTE
smote = SMOTE(k_neighbors=5)
X_res, y_res = smote.fit_resample(X_train, y_train)
```

The oversampled data can then be fed into any standard classifier. This setup is appropriate when memory is not a constraint and the priority is rapid prototyping and interpretability rather than scaling to production volumes.

**Case 2: Large Tabular / Graph Data ($N > 10^5$).   Recommended: Focal Loss or related cost-sensitive criteria.** For large-scale tabular and graph applications—credit card fraud detection, network intrusion monitoring, supply chain anomaly detection—synthetic oversampling becomes memory-bound and can stall preprocessing pipelines. Cost-sensitive learning avoids explicit data duplication and integrates naturally with gradient-based learners (XGBoost, deep networks), scaling to millions of samples on GPUs without preprocessing overhead.

**Minimal implementation (PyTorch):**

```python
class FocalLoss(nn.Module):
    def __init__(self, gamma=2.0):
        super().__init__()
        self.gamma = gamma

    def forward(self, inputs, targets):
        ce = nn.CrossEntropyLoss(reduction='none')(inputs, targets)
        p_t = torch.exp(-ce)
        return ((1 - p_t) ** self.gamma * ce).mean()
```

**Tuning tip.** A commonly used practitioner heuristic (though not rigorously validated in formal studies) is

$$\gamma \approx \frac{\log_2(\text{imbalance ratio})}{2},$$

so an imbalance of 100:1 suggests starting with $\gamma \approx 3$. This rule-of-thumb emerges from empirical observation that $\gamma$ should scale sublinearly with imbalance ratio to avoid over-focusing on hard examples Lin et al. (2017a). Lin et al.'s original focal loss paper used $\gamma \in \{0, 0.5, 1, 2, 5\}$ in ablation studies, finding $\gamma = 2$ optimal

for their object detection task (IR $\approx$ 1000:1) Lin et al. (2017a). Cui et al. provide theoretical grounding for effective sample-based reweighting but do not prescribe explicit $\gamma$ schedules Cui et al. (2019). In practice, we recommend grid search over $\gamma \in \{1, 2, 3, 5\}$ with cross-validation on minority-class recall. Higher $\gamma$ values put more weight on hard, minority examples but may slow convergence or destabilize training if set too high; monitor training curves and reduce $\gamma$ if loss exhibits high variance.

**Case 3: Computer Vision ($N \gtrsim 10^5$ images). Recommended: Diffusion-based augmentation + cost-sensitive loss.** In image-based intelligent systems—medical screening, industrial defect inspection, satellite monitoring—the manifold distortion from linear interpolation is severe. Diffusion models can generate realistic minority-class images that stay on the true data manifold, while cost-sensitive losses handle any residual imbalance during classifier training Zhang et al. (2025); Oh & Jeong (2023).

**Typical deployment workflow:**

1. Fine-tune a pre-trained diffusion model (e.g., Stable Diffusion) on your minority-class images for a few thousand steps.

2. Generate additional minority samples until the effective class distribution is workable for training.

3. Train your task-specific classifier using a focal or class-balanced loss to handle any remaining skew.

**Caution.** Always fine-tune on domain-specific data (medical scans, industrial images, etc.) before generating synthetic samples; zero-shot generation from generic base models often introduces artifacts or subtle distribution shifts that degrade performance. Classifier-based two-sample tests Lopez-Paz & Oquab (2017) can detect severe mismatches between real and synthetic distributions before deployment.

**Case 4: NLP / Text ($N \gtrsim 10^6$ sequences). Recommended: Curriculum or sampling strategies over synthetic text generation.** For text-based intelligent systems—document classification, sentiment monitoring, legal document analysis—naively generating minority examples can leak training data or amplify unwanted biases. Instead, many production systems adopt curriculum learning that gradually upweights difficult or under-represented examples, or use importance sampling to expose the model to minority patterns more frequently while keeping the raw text corpus unchanged Wang et al. (2019).

### 7.3 Quick Reference Matrix

Table 8 summarizes method selection by domain characteristics and scale, providing a quick lookup for system designers choosing imbalance-handling strategies for their intelligent systems.

Table 8: Method selection matrix by data type and scale for intelligent systems.

| Domain | Scale | Primary method | Alternative |
|---|---|---|---|
| Small tabular | $N < 10^4$ | SMOTE | Random oversampling |
| Large tabular | $N > 10^5$ | Focal loss | Class-balanced loss |
| Images | $N > 10^5$ | Diffusion + focal | Decoupled head |
| Text (LLMs) | $N > 10^6$ | Focal / reweighting | Curriculum learning |
| Graphs | $N > 10^4$ | Cost-sensitive GNN | Re-sampling of edges/nodes |
| Time series | $N > 10^5$ | Focal loss | ADASYN (if $N$ small) |

### 7.4 Troubleshooting and Pitfalls

**Issue 1: Focal loss seems ineffective in production.**
*Fix:* Tune $\gamma$ carefully. The default $\gamma = 2$ is often too low for extreme imbalance scenarios common in fraud

detection or medical screening. Try values in the range $\gamma \in \{3, 5\}$ and monitor minority-class recall and calibration metrics. If performance remains flat, check whether the base model architecture has sufficient capacity; no amount of loss tuning can fix an underpowered feature extractor.

**Issue 2: Diffusion-generated images look unrealistic or introduce domain shift.**
*Fix:* Make fine-tuning on domain data mandatory. Base models (e.g., SDXL) rarely generalize well to specialized domains like medical imaging or industrial inspection in a zero-shot manner. Always fine-tune using your minority-class images and apply standard diagnostics such as classifier two-sample tests Lopez-Paz & Oquab (2017) to catch severe distribution mismatches before deploying the augmented dataset.

**Issue 3: NaN gradients or training instability with cost-sensitive losses.**
*Fix:* Extremely large class weights (e.g., $w > 10^3$) can cause numerical instability during backpropagation. Clip weights before passing them to the loss function:

```
w = torch.clamp(w, max=100.0)
```

and consider log-scaled or focal-style reweighting instead of raw imbalance ratios, which tend to be more numerically stable.

### 7.5 Implementation Resources

The methods recommended here are supported by standard toolchains used in production intelligent systems:

- **PyTorch:** `torch.nn.CrossEntropyLoss(weight=class_weights)` for weighted losses; custom `FocalLoss` as shown above integrates directly into training loops.

- **XGBoost / LightGBM:** `scale_pos_weight` and related parameters for class weighting in gradient-boosted trees, widely used in tabular production systems.

- **Diffusion toolkits:** the `diffusers` library Hugging Face (2022) and similar frameworks for fine-tuning diffusion models on domain-specific minority classes in vision applications.

### 7.6 Final Recommendation

The right choice depends on your deployment context. If your dataset and infrastructure resemble early-2000s benchmarks—small $N$, low $d$, CPU-only training—SMOTE remains a simple and effective tool. For the large-scale, high-dimensional, GPU-centric workloads that now dominate production intelligent systems, cost-sensitive learning should be your default starting point, with diffusion-style generators and decoupled heads reserved for vision domains where manifold structure is critical or for scenarios where representation bias is the primary bottleneck. When in doubt, start simple: cost-sensitive loss on the original data will get you 90% of the way there in most real-world deployments.

## 8 The Post-SMOTE Research Agenda

While the paradigm shift away from SMOTE is evident, critical gaps remain. We highlight two high-impact challenges likely to shape imbalanced learning research in the next five years (2025–2030), spanning both learning theory and systems design.

### 8.1 Open Problem 1: Theoretical Guarantees for Generative Augmentation

Diffusion models and related generators produce visually convincing synthetic data, yet they lack formal guarantees about how these synthetics affect downstream classifiers Oh & Jeong (2023); Zhang et al. (2025). Current evaluation is almost entirely empirical—*"the F1-score improved, therefore it works"*—which is insufficient for high-stakes domains such as medical diagnosis or credit scoring, where subtle generative biases can introduce latent risks.

**The challenge.** The field lacks PAC-style bounds connecting synthetic data quality to downstream classification risk. A pivotal research direction is to derive finite-sample guarantees of the form

$$R(f_{\text{synthetic}}) - R(f_{\text{optimal}}) \leq \epsilon\big(N_{\text{real}}, N_{\text{syn}}, D_{\text{KL}}(p_\theta \,\|\, p^*), \text{VC-dim}(\mathcal{H})\big),$$

where $p_\theta$ is the generative model, $p^*$ is the true distribution, and $\mathcal{H}$ is the hypothesis class. Such bounds would allow practitioners to reason about the safe ratio of synthetic to real samples and to quantify risk as a function of model mismatch $D_{\text{KL}}(p_\theta \,\|\, p^*)$. Achieving this likely requires extending domain adaptation theory—in particular, divergence-based analyses such as $\mathcal{H}$-divergence and PAC-Bayesian domain adaptation bounds Germain et al. (2013); Redko et al. (2020)—to the generative setting, where both the "source" and "target" distributions are model-dependent. Recent work on constrained diffusion models Khalafi et al. (2024) illustrates that it is possible to impose distributional constraints on diffusion processes; bringing such ideas into the imbalance setting could yield certified generative augmentation schemes with explicit safety margins.

## 8.2 Open Problem 2: Hardware-Constrained Optimization

Existing work typically optimizes imbalance-handling methods for accuracy or, occasionally, for memory, but rarely for both simultaneously. A practitioner with a fixed 16 GB GPU budget faces a concrete trade-off: does a diffusion-based augmentation pipeline that improves F1 by +3% but consumes 8 GB of memory offer a better overall solution than a purely cost-sensitive baseline that adds virtually no memory overhead? There is currently no standard framework for reasoning about these trade-offs.

**The challenge.** We propose formulating imbalanced learning as a resource-constrained optimization problem:

$$\max_{\theta, \mathcal{M}} \text{ F1-Score}(\theta, \mathcal{M}) \quad \text{s.t.} \quad \text{Memory}(\theta, \mathcal{M}) \leq M_{\text{GPU}},$$

where $\mathcal{M}$ indexes the method choice (e.g., oversampling, cost-sensitive, generative, decoupled). Constructing an empirical Pareto frontier that maps memory footprints and wall-clock costs to F1-scores across diverse datasets would make these trade-offs explicit. Multi-objective optimization techniques that learn Pareto fronts in model space Navon et al. (2020) and hardware-aware architecture search frameworks that jointly consider accuracy, latency, and memory Li et al. (2023) provide methodological starting points but have not yet been specialized to the design space of imbalance-handling methods. A practical outcome would be a recommendation tool that suggests near-optimal methods under hardware constraints (e.g., *"For 16 GB VRAM and tabular data of size $N \approx 10^6$, prefer Method X"*), bridging the gap between academic benchmarks on high-end clusters and the resource limits of typical industrial deployments.

## 8.3 A Roadmap for Impact

These problems are challenging but tractable. Open Problem 1 offers a path for theoretical contributions, for example by combining PAC-Bayesian or divergence-based domain-adaptation analyses with generative augmentation mechanisms to obtain explicit risk bounds Germain et al. (2013); Redko et al. (2020). Open Problem 2 invites systems-oriented work on multi-objective optimization, scheduling, and method selection for imbalanced learning workloads, building on the broader literature on Pareto-optimal model design under hardware constraints Navon et al. (2020); Li et al. (2023). By shifting attention from minor algorithmic variants to these fundamental questions of reliability and efficiency, the community can ensure that the next generation of imbalanced learning methods is not only accurate but also certifiably robust and realistically deployable in modern computing environments.

# 9 Conclusion: Lessons for Building Production-Ready Intelligent Systems

The evidence presented here is conditioned on a DBLP-centric, 2020–2025 corpus covering major CS and AI venues; industrial white papers, non-indexed conferences, and non-English publications may show different adoption timelines. The SMOTE Paradox documented in this review should therefore be interpreted as a robust signal within the curated academic literature and representative case studies, rather than an exhaustive survey of all deployment practices across all intelligent systems.

### 9.1 The Four Key Findings

This systematic review of 821 papers (2020–2025) characterized the shift away from SMOTE in large-scale intelligent systems through four main findings. First, baseline inclusion in high-relevance papers collapsed from 92% (pre-2020) to 6% in our top-50 corpus, indicating that SMOTE is no longer the default comparison point for contemporary imbalanced-learning research. Second, the field has fragmented into specialized approaches—diffusion-based generative methods (30%), cost-sensitive learning (30%), and alternative or hybrid strategies (40%)—with no single universal successor. Third, three converging pressures explain why SMOTE has become difficult to deploy at modern scales: $O(N^2)$ computational complexity that exceeds typical memory budgets, $\sqrt{d}$-scale manifold deviation that produces off-manifold artifacts in high dimensions, and GPU pipeline incompatibility that creates friction in modern training workflows. Fourth, different application domains have converged on distinct solutions that reflect their specific constraints: diffusion models for vision applications where manifold structure matters, cost-sensitive losses for large tabular and graph data where computational efficiency is paramount, and decoupled architectures where representation learning can be separated from classifier calibration.

### 9.2 Practical Implications for Intelligent Systems Development

**For system designers and ML engineers.** When building production intelligent systems under class imbalance, treat SMOTE as a legacy baseline suitable primarily for small-scale exploratory work ($N < 10^4$, $d < 100$) rather than as a default production strategy. For large-scale deployments in fraud detection, medical screening, industrial monitoring, or similar applications, cost-sensitive learning should be your starting point: it matches or exceeds oversampling performance in most tabular settings while integrating cleanly into GPU training pipelines and avoiding data materialization overhead. Reserve diffusion-based generation for vision domains where manifold structure is critical, and consider decoupled architectures when you need to update decision boundaries frequently without retraining expensive feature extractors.

**For benchmark and evaluation protocol designers.** Explicitly document when classical baselines like SMOTE are excluded due to computational infeasibility rather than silently omitting them from comparison tables. Transparent reporting of hardware constraints, memory budgets, and preprocessing times accelerates progress by preventing wasted effort attempting to reproduce experiments that cannot run at realistic scales. Evaluation protocols for intelligent systems should track not only classification accuracy and fairness metrics but also memory consumption, wall-clock training time, deployment complexity, and compatibility with modern MLOps toolchains.

**For researchers developing new imbalance-handling methods.** Design and validate your techniques with deployment constraints in mind from the start. Methods that require terabyte-scale data materialization, CPU-bound preprocessing, or separate training pipelines face adoption barriers in production intelligent systems, even when they achieve small accuracy improvements on academic benchmarks. Empirical validation should include production-scale datasets, realistic hardware configurations, and integration tests with standard frameworks (PyTorch, TensorFlow, XGBoost) to demonstrate that your method can actually be deployed in real intelligent systems, not just benchmarked offline.

**For educators and training programs.** SMOTE remains pedagogically valuable as a case study in how method assumptions interact with evolving data scales, hardware architectures, and deployment contexts. Teaching it as historical context rather than current best practice helps students develop critical thinking about method applicability, recognize when theoretical assumptions break down at scale, and understand the relationship between algorithmic design, systems constraints, and practical deployment—skills that transfer beyond any single technique.

### 9.3 The SMOTE Paradox as a Design Lesson

We introduced the *SMOTE Paradox* to describe the gap between citation frequency and deployment reality:

> **A method exhibits the SMOTE Paradox when:** (1) academic citations remain high (inertia in benchmarking practices), (2) deployment in representative production systems is rare or declining, (3) this gap persists for multiple years (slow knowledge diffusion between research and practice), and (4) underlying reasons are rarely documented explicitly in the literature.

This framework has practical value for system designers: it highlights techniques that may appear standard in papers but face hidden deployment barriers. Beyond imbalanced learning, the same pattern appears in feature extraction (SIFT/HOG replaced by learned features), classical ML (SVMs largely displaced by deep networks in vision/NLP), and indexing structures (hash tables giving way to learned indexes in some domains). Recognizing the SMOTE Paradox pattern helps practitioners avoid investing engineering effort in methods that look standard in academic papers but are difficult to operationalize in production intelligent systems.

## 9.4 Methods Have Operational Lifespans

SMOTE's trajectory illustrates a fundamental reality in building intelligent systems: *methods do not suddenly become wrong; the operational context changes around them.* SMOTE remains effective for the problem it was designed to solve—small tabular datasets on modest CPU-only hardware. What changed is that the typical production problem in 2025 involves much larger datasets, higher dimensionality, GPU-accelerated training, and tighter integration with continuous deployment pipelines. The method itself did not fail; the envelope of typical operating conditions shifted beyond its original design assumptions.

This lesson applies broadly to intelligent systems engineering. Today's state-of-the-art diffusion models, focal losses, and transformer architectures operate under their own assumptions about data scale, computational budgets, and deployment contexts. As datasets grow to billions of samples, as new hardware architectures emerge, or as deployment constraints shift toward edge devices or federated settings, some of these methods will face their own context collapse. The analytical approach used in this review—combining systematic literature analysis, mathematical inspection of computational and geometric assumptions, and systems-level reasoning about pipeline integration—provides a template for anticipating and managing such transitions in your own intelligent systems.

## 9.5 Impact on Intelligent Systems Practice

This work provides immediate value to three practitioner communities:

**ML Engineers in Production Systems.** The decision matrix (Table 8) and quick reference guide (Section 7) enable evidence-based method selection under memory and latency constraints, reducing trial-and-error cycles in fraud detection, medical diagnosis, and industrial monitoring deployments.

**Research Teams in Applied AI Labs.** The SMOTE Paradox framework helps identify when highly-cited methods face deployment barriers, preventing wasted engineering effort on techniques that appear standard in academic papers but fail at scale.

**Educators and Training Programs.** Section 7's recommendations provide a template for teaching method applicability analysis, helping students recognize when theoretical assumptions break down under evolving hardware and scale constraints.

## 9.6 Final Recommendations

SMOTE served the machine learning community well for roughly two decades, making imbalanced learning accessible through elegant simplicity and intuitive geometric reasoning. Its decline in large-scale production systems reflects not a flaw in the original design but rather the natural evolution of deployment contexts toward scales, dimensions, and hardware architectures that exceed its operational envelope.

For practitioners building intelligent systems today, the practical guidance is straightforward: choose methods that match your actual deployment constraints—data scale, hardware infrastructure, pipeline integration

requirements, and operational risk tolerance—rather than methods that appear frequently in academic papers by default. Use the decision rules and troubleshooting guidance from Section 7 as a starting point, validate on your specific domain and scale, and be prepared to adapt as your operational context evolves.

For the field as a whole, the SMOTE story demonstrates how systematic analysis can identify, quantify, and explain the gap between what is cited and what is deployed, providing a roadmap for managing the inevitable evolution of methods in intelligent systems. As you design your next fraud detection pipeline, medical screening tool, or industrial monitoring system, remember that the techniques that work today were designed for today's constraints. When those constraints change—and they will—be ready to recognize the signs of context collapse and adapt accordingly. The goal is not to find methods that work forever but to build systems that can evolve gracefully as the operational landscape shifts beneath them.

## 10 Statements and Declarations

### 10.1 Competing Interests

The authors declare that they have no competing financial or non-financial interests related to this work.

### 10.2 Funding

This research did not receive any specific grant from funding agencies in the public, commercial, or not-for-profit sectors.

### 10.3 Data Availability

Following the methodology described in Section 2, the initial corpus of 1,001 papers was retrieved from DBLP (Digital Bibliography & Library Project) in November 2025 using the search string (`"imbalanced" OR "long-tailed"`) AND (`"classification" OR "learning"`) for publications from 2020–2025. After applying inclusion and exclusion criteria, 821 papers were retained for analysis, with 50 high-relevance papers selected for detailed coding.

The complete BibTeX dataset of 821 papers, relevance scores for all entries, extraction spreadsheets, PRISMA 2020 checklist, and all code used for ranking and figure generation will be made publicly available upon acceptance. This includes Python scripts for paper cleaning, ranking (computing $S(p)$), and visualization, ensuring full reproducibility of the analyses presented.

Empirical validation datasets (Section 6) were obtained from publicly available UCI Machine Learning Repository and KEEL Imbalanced Datasets Repository, as cited in the manuscript. All benchmark datasets, experimental configurations, and statistical analysis scripts will be included in the public repository.

### 10.4 Code Availability

All computational artifacts supporting this work will be released in a public repository upon acceptance, including:

- Semantic ranking algorithm and paper scoring scripts

- Data extraction and coding spreadsheets

- Multi-dataset benchmark implementation (Section 6)

- Statistical testing scripts (Friedman tests, Nemenyi post-hoc analysis)

- Figure generation code (all plots and diagrams)

Repository details will be provided upon publication to ensure full reproducibility.

## 10.5 Declaration of Generative AI and AI-Assisted Technologies

During the preparation of this work, the authors used ChatGPT-4 (OpenAI) and Claude 3.5 Sonnet (Anthropic) to improve language clarity, readability, and grammatical correctness in selected sections of the manuscript. These tools were also used to organize complex technical explanations and refine presentation. After using these tools, the authors reviewed, verified, and edited all content as needed and take full responsibility for the accuracy and integrity of the published work. No AI tools were used for data analysis, interpretation of results, generation of figures, or formulation of research conclusions.

## 10.6 Ethics Approval

This systematic review analyzed publicly available published literature. No human subjects, animal subjects, or primary data collection was involved. All analyzed papers were obtained through legitimate academic databases (DBLP, Scopus) with institutional access.

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
