# OpenReview forum: "The SMOTE Paradox: Why a 92% Baseline Collapsed to 6%—A Systematic Review of 821 Papers in Imbalanced Learning (2020–2025)"
_TMLR — Rejected by TMLR_

### Review · Reviewer_5cZQ · 2026-01-20

**Summary Of Contributions:**

The paper presents a large-scale systematic review and bibliometric analysis of imbalanced learning literature from 2020–2025, focusing on the declining practical use of SMOTE despite its continued prominence in citations. By analyzing 821 papers and conducting controlled experiments on several benchmark datasets, the authors introduce the concept of the “SMOTE Paradox”: SMOTE remains widely cited but is rarely used successfully at scale. The paper attributes this decline to computational constraints, geometric limitations in high-dimensional spaces, and incompatibility with modern GPU-centric training pipelines. It further categorizes post-SMOTE alternatives (e.g., cost-sensitive learning and generative methods) and provides high-level recommendations on when different imbalance-handling strategies should be used, depending on dataset size and imbalance ratio.

Key Strengths:
1. The paper surveys a large number of recent studies (821 papers), with clear effort devoted to systematic review methodology and bibliometric analysis.
2. The paper effectively highlights the gap between SMOTE’s citation popularity and its limited deployment in modern research and practice.
3. The discussion of memory and scalability issues provides a practical perspective often missing from imbalance-learning surveys.

Key Weakness:
1. Limited conceptual novelty in the core explanation of SMOTE’s limitations
2. A reliance on experimental settings that may not fully reflect modern deep learning applications

**Audience:**

No

**Audience Explanation:**

Although the topic of imbalanced learning is relevant, the main findings—that SMOTE does not scale well to modern settings due to computational, geometric, and pipeline constraints—are largely intuitive and already well understood by researchers in this area. As a result, the paper offers limited new insight beyond what many readers are likely to already know.

While the large-scale literature survey may be informative for newcomers, the lack of conceptual novelty and the reliance on small, traditional tabular benchmarks reduce the paper’s appeal for a broader TMLR audience, particularly those working on modern deep learning systems and large-scale applications.

**Broader Impact Concerns:**

One potential concern is that the paper’s recommendations—particularly regarding the use of SMOTE for small datasets—could be misinterpreted or overgeneralized by practitioners, leading to suboptimal or misleading model performance in sensitive domains such as healthcare or finance if imbalance-sensitive metrics and threshold selection are not carefully considered. However, this risk is indirect and could be mitigated through clearer caveats and more precise guidance.

**Claims And Evidence:**

No

**Claims Explanation:**

The claims are partially supported, but the evidence is not consistently convincing or well aligned with the paper’s broader conclusions.

The literature review and bibliometric analysis convincingly support the existence of a citation–deployment gap for SMOTE, and the discussion of computational and scalability limitations is accurate and broadly consistent with practitioner experience.

However, several specific claims lack strong empirical support:

1. The takeaway that SMOTE does not scale well to modern settings due to the main three claims listed in the paper is too intuitive, reducing the conceptual novelty.
2. Figure 4 is used to illustrate the publication trend of post-SMOTE methods; however, the change in counts from 8 to 16 and then down to 11 is too small and inconsistent to convincingly support the claim that the emergence of new imbalance-handling methods is driving a meaningful shift in the popularity of post-SMOTE research.
3. Despite framing the discussion around modern AI/ML systems, all experiments are conducted on UCI-style tabular datasets with traditional machine learning models, which weakens the relevance to large-scale vision, text, or deep learning scenarios.
4. The evaluation discussion focuses heavily on AUROC, where most methods appear similar, while imbalance-sensitive metrics such as AUPRC and F1—often more relevant in practice—receive less emphasis.
5. Threshold-dependent metrics are reported, but the rationale for threshold selection and its influence on the reported conclusions are not sufficiently analyzed. In particular, the use of a fixed threshold of 0.5 can be problematic in imbalanced classification settings, where such a choice is rarely optimal.
6. The paper concludes that SMOTE can be recommended for small datasets; however, the reported results show that, when F1 score is considered, the no-resampling baseline often performs best, and SMOTE can even degrade performance. This undermines the strength of the recommendation.

**Requested Changes:**

1. To support claims about modern AI/ML systems and GPU-centric pipelines, the authors should include experiments on large-scale or high-dimensional datasets (e.g., vision, text, or deep learning benchmarks) rather than relying solely on small UCI-style tabular datasets with traditional models.

2. The evaluation should place greater emphasis on imbalance-sensitive metrics such as AUPRC and F1, rather than focusing primarily on AUROC. The practical implications of metric choice should be discussed more explicitly.

3. For threshold-dependent metrics, the authors should justify how thresholds are chosen and analyze the sensitivity of results to different thresholds. Relying on a fixed threshold of 0.5 is inappropriate for imbalanced classification and may bias conclusions.

4. The recommendation that SMOTE is suitable for small datasets should be re-evaluated in light of the reported F1 results, which often favor the no-resampling baseline and sometimes show performance degradation with SMOTE.

5. The authors should more clearly articulate what is new beyond widely accepted intuitions about SMOTE’s limitations, or temper claims of novelty accordingly.

---

### Review · Reviewer_PHUJ · 2026-02-02

**Summary Of Contributions:**

This paper is primarily a survey paper covering SMOTE (a technique for dealing with classifiers trained on datasets with class imbalance). The paper points out that SMOTE was previously highly popular, but they find that in the corpora of papers selected for their survey, only 6% compare to SMOTE as a baseline. The paper proposes a few reasons for this disconnect, mainly focusing on the challenging scalability of the $k$-nearest-neighbor lookup that typical implementations of SMOTE use. Some experiments are run to examine the performance of SMOTE on datasets of varying sizes.

**Audience:**

No

**Audience Explanation:**

I think there are too many un-justified claims that underlie the findings for the findings to be of interest to the TMLR audience. I'm particularly concerned about the 6% number being heavily influenced by the chosen methodology, rather than an actual full survey of the (applied) ML literature showing a decrease in the use of SMOTE (see the final bullet point above).

**Broader Impact Concerns:**

I don't think this work raises any ethical concerns.

**Claims And Evidence:**

No

**Claims Explanation:**

I have a few issues with the claims made in the paper; I've broken these down into three sections below.

# Direct claims made without evidence
- Section 1.1 breaks down approaches to imbalance handling into three "generations." First, Generation I starts in 2000, but this isn't the start of this field; see the related work section of Chawla et al. (2002) (the SMOTE paper) for examples. Second, this whole section needs more references. E.g., I don't think one can define a whole generation and then provide only a single example method that fits that generation (which is how Generation II currently is)
- "practitioners report abandoning SMOTE in large-scale deployments due to memory exhaustion, slow preprocessing, and incompatibility with GPU-centric MLOps pipelines" -- this is a central point of the paper (that practitioners are abandoning SMOTE), but no evidence is provided.
- "Analysis of recent Kaggle competitions and production ML systems shows that winning solutions for million-sample image or text tasks rarely employ SMOTE as their primary balancing strategy" -- no evidence is provided.
- "Production teams consistently report out-of-memory crashes, preprocessing delays, and integration diﬃculties when attempting to incorporate SMOTE-style oversampling into modern GPU training loops." -- no evidence is provided.
- "For a manifold... with non-negligible curvature, the expected distance from a linear interpolant ... [to the manifold] ... grows with dimension [$\sqrt{d}]" -- no evidence is provided.
- "Generative methods that approximate the score or density of the data distribution ... stay on or near the true data manifold" -- no evidence is provided.
- "Methods that preserve the original dataset and act within the loss function or classiﬁer head are easier to integrate into mature MLOps stacks," -- I don't think the above arguments show this, as the above arguments haven't referenced any specific MLOps stacks to show that such methods easily integrate into said stacks.
- "A practical rule of thumb [for tuning $\gamma$]..." this doesn't seem to be based on anything.
- The paper claims that SMOTE is also harder to integrate into existing ML pipelines because it modifies the dataset. I have two thoughts on this. First, the paper notes the increasing popularity of diffusion based methods for data augmentation, which also modify the dataset (by adding to it, just as SMOTE does), so I don't think this is an explanation for less-common use of SMOTE. Second, I don't believe that the paper has sufficiently made the point that other dataset balancing methods are easy to integrate; the paper just states this as a fact: "[b]ecause [other methods] do not alter raw data, they remain compatible with existing data validation tools, schema evolution frameworks, and feature-store infrastructure used in production intelligent systems."
- Section 4.3 implies that SMOTE is not commonly used in part because it is not compatible with GPUs. But there's a paper from 2017 about running SMOTE on the GPU (Gutiérrez et al. 2017, "SMOTE-GPU: Big Data preprocessing on commodity hardware for imbalanced classification"

# Experiments
- This paper is about SMOTE failing on modern datasets with modern ML methods. The experiments then only use UCI datasets and use XGBoost, Random forests, LightGBM and logistic regression. I really think entirely new experiments are needed -- that use truly modern datasets and models -- to back up the claims of the paper.
- Fig 10 shows the performance of various resampling methods across these experiments. But the performance of No Resampling is almost just as good as every other method. This seems to indicate that these dataset/model combinations don't really need resampling to work well. Focusing on dataset/model combinations with bad baseline performance would better draw out the differences between the resampling methods.
- "Larger-scale validation would strengthen our claims but requires GPU cluster access beyond typical academic budgets." I don't think this claim is true. E.g., H200 GPUs are available for rent for less than $10/hour: https://www.hyperstack.cloud/gpu-pricing?utm_medium=organic&utm_source=google (each of which has *VRAM* equal to 7x the RAM of the computer used for experiments here).
- Relatedly, Section 2.6 states that the hardware used (a workstation with 20 GB of RAM and a GTX1650 GPU) reflects "realistic [consumer-grade] hardware constraints discussed in Section 4.1." First, I disagree that this is realistic hardware -- e.g. I have a five year old personal desktop that was <$1,500 at the time, and it's more powerful than this machine. Second, the only place I saw a discussion of hardware in Section 4.1 was to say that at least 64GB of RAM is typical, which is also more than this machine.

# Methodology
The methodology for selecting papers to be included in the survey is precisely defined but isn't particularly well-motivated:

- One part of the survey was done in November 2025 and the other in January 2026. The paper argues "The two-month temporal gap between database queries is negligible given our focus on multi-year trends". If this is the case, why have a difference between these two dates at all?
- Papers are ranked based on keywords present in their titles, with different weights attached to each keyword. First, it's not clear to me that the title is the best place to look -- it seems like the abstracts plus titles might be better. Second, there's no description of how these keywords were chosen or how these weights were assigned.
- Using the keyword weighting approach, the paper narrows down the papers to just 50 highly-ranked papers. It finds that only 3 of the papers use SMOTE as a baseline, and none of these papers could "execute [SMOTE] at full scale due to memory overflow." I would argue that this finding is heavily influenced by the chosen keywords. In particular, the keywords include things like "scalability, OOM, memory, computational cost". Of course the papers chosen are going to find SMOTE infeasible to run -- they've been specifically selected to find papers about scalability issues! So I really don't think the "6% of papers" (3/50) claim is accurate; this number just shows that, out of papers that are about scalability and dataset imbalance, only 3/50 of the papers actually attempt to run SMOTE (presumably because of scalability issues).

**Requested Changes:**

# Critical for acceptance
- Improve methodology or better argue why the current methodology isn't highly biasing the results
- Make sure all claims have either evidence from the results in the paper *or* have references to back them up
- Include much larger-scale experiments with truly modern ML models and datasets (e.g., by renting more performant servers for experiments)
- Fix the typos present in the paper; there are a lot of typos, particularly with words running together (e.g., "environmentswhere"). I counted four typos like this on the first page of the paper.
- Fix the formatting issue on p.4 causing a huge amount of blank space

---

### Review · Reviewer_PWg5 · 2026-04-15

**Summary Of Contributions:**

The paper illustrates and explains the decline in SMOTE's role in imbalanced learning through a systematic review of 821 DBLP papers from 2020 to 2025, a bibliometric analysis of 4,985 Scopus records, and a 196-trial benchmark across seven tabular datasets. The authors explain the "SMOTE Paradox": while 24% of papers mention SMOTE, only 6% of "production-oriented" papers successfully execute it. It is mainly due to three failure modes: O(N * Nmin * d) complexity, a sqrt(d) manifold distortion, and CPU-GPU pipeline friction. A Friedman test on the benchmark finds no significant difference in ROC-AUC between SMOTE and cost-sensitive baselines. The paper concludes with a decision matrix for selecting methods. The contribution is not methodological novelty but rigorous quantification of a well known but undocumented dilemma in the ML community.

**Additional Comments:**

This is a solid contribution that provides systematic evidence of what the ML community has suspected for a long time. I want to highlight one finding the authors could emphasize more: the statistical equivalence of SMOTE and doing nothing (p = 0.907) on small tabular data, where SMOTE was designed to work, may be more damaging than the scalability argument, because it means SMOTE adds no value even in its best-case scenario

**Audience:**

Yes

**Audience Explanation:**

This paper tackles a real gap in the literature. Many practitioners and researchers are aware of SMOTE's decline, but no prior work has systematically documented it with statistical support and a bibliometric framework. The finding that classifier choice yields 7-9% performance gains, while resampling differences are 1-2%, is an actionable result. The decision matrix in Section 7 is very useful for anyone building imbalanced classification systems in production.

**Claims And Evidence:**

Yes

**Claims Explanation:**

The three-way triangulation (bibliometric, theoretical, empirical) is relevant, and each pillar reinforces the others. The computational complexity analysis (Section 4.1) is clearly derived. The PRISMA workflow includes robustness checks and expert validation. The benchmark follows Demsar's statistical methodology, and the critical difference diagram (Figure 9) supports the equivalence claim.

**One concern**

ROC-AUC as the primary metric is a questionable choice for a paper about imbalanced classification. ROC-AUC is not sensitive to class imbalance. The authors themselves note that G-mean reveals much more method separation (resampling: 0.81-0.84 vs. NoResampling: 0.67, Table 5). PR-AUC may have been more appropriate.

**Requested Changes:**

1. [major] Report PR-AUC (or Average Precision) alongside ROC-AUC. In a paper on imbalanced classification, the primary metric should be sensitive to the minority class's performance. If statistical equivalence holds under PR-AUC too, it strengthens the paper considerably. If not, that is also worth discussing.

This could strengthen the paper:

1. [minor] Add a brief background section covering SMOTE, its variants, and modern alternatives. These explanations are spread through several sections.
2. [minor] The paper is long for its format. Sections 5 (MLOps implications) and parts of Section 7 read more like a tutorial than a research contribution.

---

### Decision · Action_Editor_Eb9c · 2026-05-30

**Recommendation:** Reject

**Audience:**

Yes

**Audience Explanation:**

Handling data imbalance is critical when dealing with real-world data across many applied domains of machine learning.

**Claims And Evidence:**

No

**Claims Explanation:**

As pointed out by multiple reviewers, the models, evaluation metrics, and datasets used in the empirical study are highly limited. In addition, reviewer PHUJ has highlighted numerous claims in the paper without any evidence